# Structure-guided functional suppression of AML-associated DNMT3A hotspot mutations

Jiuwei Lu [1,7], Yiran Guo [2,3,7], Jiekai Yin[4], Jianbin Chen[1], Yinsheng Wang[4,5], Gang Greg Wang [2,3,6] ✉ & Jikui Song [1,4] ✉

DNA methyltransferases DNMT3A- and DNMT3B-mediated DNA methylation critically regulate epigenomic and transcriptomic patterning during development. The hotspot DNMT3A mutations at the site of Arg822 (R882) promote polymerization, leading to aberrant DNA methylation that may contribute to the pathogenesis of acute myeloid leukemia (AML). However, the molecular basis underlying the mutation-induced functional misregulation of DNMT3A remains unclear. Here, we report the crystal structures of the DNMT3A methyltransferase domain, revealing a molecular basis for its oligomerization behavior distinct to DNMT3B, and the enhanced intermolecular contacts caused by the R882H or R882C mutation. Our biochemical, cellular, and genomic DNA methylation analyses demonstrate that introducing the DNMT3B-converting mutations inhibits the R882H-/R882C-triggered DNMT3A polymerization and enhances substrate access, thereby eliminating the dominant-negative effect of the DNMT3A R882 mutations in cells. Together, this study provides mechanistic insights into DNMT3A R882 mutations-triggered aberrant oligomerization and DNA hypomethylation in AML, with important implications in cancer therapy.

Cytosine-5 DNA methylation is an evolutionarily conserved epigenetic mechanism that is essential for silencing of retrotransposons[1,2], X-chromosome inactivation[3], genome imprinting[4], and cell fate determination[5,6]. In mammals, establishment of DNA methylation is mainly achieved by de novo DNA methyltransferases DNMT3A and DNMT3B[7], which bear a similar C-terminal methyltransferase (MTase) domain preceded by two histone reader modules: a proline–tryptophan–tryptophan–proline (PWWP) domain and an ATRX–DNMT3–DNMT3L-type (ADD) zinc finger domain[8,9]. It has been established that DNMT3A and DNMT3B possess overlapped but distinct functionalities[7,10,11], underpinned by their distinct stage- and tissue-specific expression patterns in development[7,12–14], as well as their divergent enzymatic properties[15,16]. In line with this notion, individual knockout (KO) of Dnmt3a or Dnmt3b led to postnatal (for Dnmt3a) or embryonic (for Dnmt3b) lethality of mice[7]. Furthermore, mutation of

DNMT3A has been linked to acute myeloid leukemia (AML)[17] and Tatton-Brown–Rahman syndrome[18], whereas mutation of DNMT3B is mainly linked to immunodeficiency, centromeric instability, facial anomalies syndrome[7,19–21].

DNMT3A-mediated DNA methylation in germ cells is critically regulated by DNMT3-like protein (DNMT3L)[22–24]. Structural studies of DNMT3A MTase domain in complex with DNMT3L C-terminal domain reveal a tetrameric assembly where DNMT3A homodimerizes via a hydrophilic (a.k.a. RD) interface and each of the DNMT3A monomer further associates with DNMT3L via a hydrophobic (a.k.a. FF) interface[25–27]. Such a tetrameric assembly was similarly observed for the complex of DNMT3A with DNMT3B isoform 3 (DNMT3B3)[28,29], and the DNMT3B–DNMT3L complex[15,30]. The structural integrity of both the RD and FF interfaces is important for DNMT3A function, as disruption of either led to impairment of enzymatic activity and/or

[1]Department of Biochemistry, University of California, Riverside, CA, USA. [2]Department of Pharmacology and Cancer Biology, Duke University School of Medicine, Durham, NC, USA. [3]Duke Cancer Institute, Duke University School of Medicine, Durham, NC, USA. [4]Environmental Toxicology Graduate Program, University of California, Riverside, CA, USA. [5]Department of Chemistry, University of California, Riverside, CA, USA. [6]Department of Pathology, Duke University School of Medicine, Durham, NC, USA. [7]These authors contributed equally: Jiuwei Lu, Yiran Guo. ✉e-mail: greg.wang@duke.edu; jikui.song@ucr.edu

chromatin targeting of DNMT3A[26,31]. Our previous structural studies of DNMT3A–DNMT3L and DNMT3B–DNMT3L in complex with CpG DNA further reveal that the substrate binding of DNMT3A and DNMT3B is primarily mediated by a loop from the target recognition domain (TRD) that contacts the DNA major groove, the catalytic loop that contacts the DNA minor groove, and an α-helix at the RD interface (RD-helix) that interacts with DNA backbone[15,27,32]. DNMT3A and DNMT3B differ in sequence and structure at the TRD loop and the RD-helix, which underpins their distinct CpG-recognition[33]. The fact that the RD interface mediates both DNMT3A/DNMT3B oligomerization and their substrate bindings highlights the essential role of the RD interface in DNMT3A/DNMT3B function.

In the absence of regulatory factor DNMT3L/DNMT3B3, DNMT3A, and DNMT3B have been shown to exist dominantly in homotetrameric forms in vitro[15,31,34,35], which might interact with DNA to form filament-like macro-oligomers[31,33]. To illuminate the mechanism of DNMT3B homo-oligomerization, we recently solved the crystal structure of the MTase domain of DNMT3B, which assumes a helical polymer in crystals via alternating RD and FF interfaces[33], as well as the cryo-EM structures of DNMT3B PWWP-ADD-MTase tetramer and other oligomers with ascending or descending order (e.g., trimer or hexamer)[36]. Structural analysis of these complexes reveals a coupling between the formation of the RD interface and the folding of the underlying structural elements, comprised of the TRD and S-adenosyl methionine (SAM)-binding site[36], establishing an intricate link between DNMT3B oligomerization, RD interface conformation and substrate binding. However, due to a lack of structure for homo-oligomeric DNMT3A, the molecular basis for DNMT3A homo-oligomerization remains elusive.

Mutation of DNMT3A has been identified in ~15–20% of leukemia patients, with the R882 mutations (in an occurrence order of R882H > R882C > R882P, R882S) accounting for over 50% of all missense mutations[37]. It has been demonstrated that the R882 mutations led to attenuated DNA methylation by WT DNMT3A in mouse ES[38] or TF1 leukemia cell line[39], suggesting a dominant-negative effect of these mutations. Consistently, heterozygous R882 mutations in normal karyotype AML patients are associated with a more pronounced reduction of DNA methylation than the rest of DNMT3A mutations[40], and R882H mutation in individuals with clonal hematopoiesis similarly led to DNA hypomethylation at selective loci[41]. Furthermore, co-expression of DNMT3A R882H mutant with WT led to increased macro-oligomer formation but compromised DNA methylation in vitro and in cells[40,42]; introducing a secondary mutation into DNMT3A R882H, which disrupts the FF interface, greatly reduced the dominant-negative effect of DNMT3A R882H mutant in cells[39]. These observations suggest a link between the dominant-negative activity and the polymerization-promoting behavior of DNMT3A R882H mutation. In addition, previous studies from others and us have demonstrated that the DNMT3A R882H mutation leads to reduced enzymatic activity[27,32,40,42–44], altered substrate preference[32,45,46], and abnormal protein interactions[47–49] of DNMT3A. However, due to the lack of a clear-cut mechanistic understanding, how these aberrant structural and biochemical properties of DNMT3A R882 mutants collectively contribute to the aberrant DNA methylation and dysregulated gene expression in AML and other development disorders remains controversial[38–40,43,50].

To illustrate the structure and mechanism of DNMT3A homo-tetramer and the pathological consequence of DNMT3A R882 mutations, here, we solved the crystal structure of DNMT3A MTase domain, revealing a homo-tetrameric assembly resembling that of the DNMT3A–DNMT3L complex. Structural comparison between the internal and external subunits of DNMT3A homotetramer reveals that, similar to DNMT3B, DNMT3A undergoes a disorder-to-order transition upon the RD interface-mediated oligomerization, indicating an oligomerization-coupled protein folding. Furthermore, our combined structural and biochemical analysis of DNMT3A and DNMT3B reveals

that amino acid variations within the RD interfaces of DNMT3A and DNMT3B give rise to their different protein oligomerization behaviors, with the DNMT3A R882 mutant being more prone to high-order oligomer formation, i.e., polymerization than the corresponding DNMT3B mutant. Guided by these observations, we introduced a "DNMT3B-converting" mutation, DNMT3A R676K to DNMT3A R882H or R882C mutant, which permits us to solve the structure of R882H- or R882C-mutated DNMT3A homo-tetramer. The structures of R882H- and R882C-mutated DNMT3A reveal that the replacement of a solvent-exposed, flexible arginine by a smaller amino acid led to enhanced intermolecular contacts at the RD interface, thereby providing an explanation for the polymerization-promoting effect of these mutations. Finally, our genomic DNA methylation analysis demonstrates that the DNMT3B-converting mutations greatly dampened the dominant-negative effect of DNMT3A mutations. Together, our studies provide a basis for mechanistic understanding and inhibition of the dominant-negative effect of DNMT3A R882 mutations, shedding light on the development of an effective therapeutic strategy against AML and other DNMT3A R882 mutation-associated diseases.

## Results

### Crystal structure of DNMT3A homotetramer reveals partially disordered external subunits

To provide the structural basis for the homotetrameric assembly of DNMT3A, we crystalized the MTase domain of DNMT3A (residues 628–912) with the SAM analog S-adenosyl-homocysteine (SAH) and solved the crystal structure at 3.3 Å resolution (Supplementary Table 1). The DNMT3A–SAH complex was packed in the P3 space group, with each asymmetric unit containing two DNMT3A tetramers (Fig. 1a, Supplementary Fig. 1a, and Supplementary Table 1). DNMT3A homotetramer assumes a similar assembly fashion as that of the previously reported DNMT3A–DNMT3L complex[25–27,32], with two central subunits dimerizing via the RD interface and each of the central subunit further associating with one external subunit. As a result, the two central subunits are flanked by both RD and FF interfaces, whereas the external subunits have a disrupted RD interface (Fig. 1a).

For the two central subunits, we were able to trace nearly the entire DNMT3A MTase domain, except for the TRD loop (residues 833–846) that reportedly undergoes a disorder-to-order transition upon DNA binding[25–27,32]. In contrast, a larger extent of structural disorder was observed for the two external subunits, with residues 671–678, 710–724, 782–789, and 805–879 disordered or completely untraceable (Fig. 1a). Among these, residues 710–724 constitute the catalytic loop, residues 671–678-situated α2-helix is part of the SAM-binding pocket, and residues 805–879 encompass the DNA-binding TRD (Fig. 1b, c)[27]. Accordingly, the SAH molecules are only bound to the two central subunits, but not the two external subunits (Fig. 1a).

### Oligomerization-coupled folding of the RD interface in DNMT3A and DNMT3B

Structural analysis of the DNMT3A RD interface reveals that it is mainly formed by two structural elements: α2-helix and TRD (Fig. 1a). Note that these regions are involved in little intermolecular contact in crystals, suggesting that their structural disorder in the external subunits is unlikely due to a crystallization packing effect (Supplementary Fig. 1b). In this context, the observation that the TRD and SAM-binding pocket assume structural order in the RD interface-present central subunits but become disordered in the RD interface-lacking external subunits (Supplementary Fig. 1b–d) implies a coupling between folding of these structural elements and integrity of the RD interface. In support of this notion, our recent structural characterization of DNMT3B homotetramer by cryo-electron microscopy (cryo-EM) reveals a similar assembly pattern (Fig. 1d, e and Supplementary Fig. 1c), with the TRD and SAM-binding pocket structurally ordered for the central subunits but disordered for the external subunits

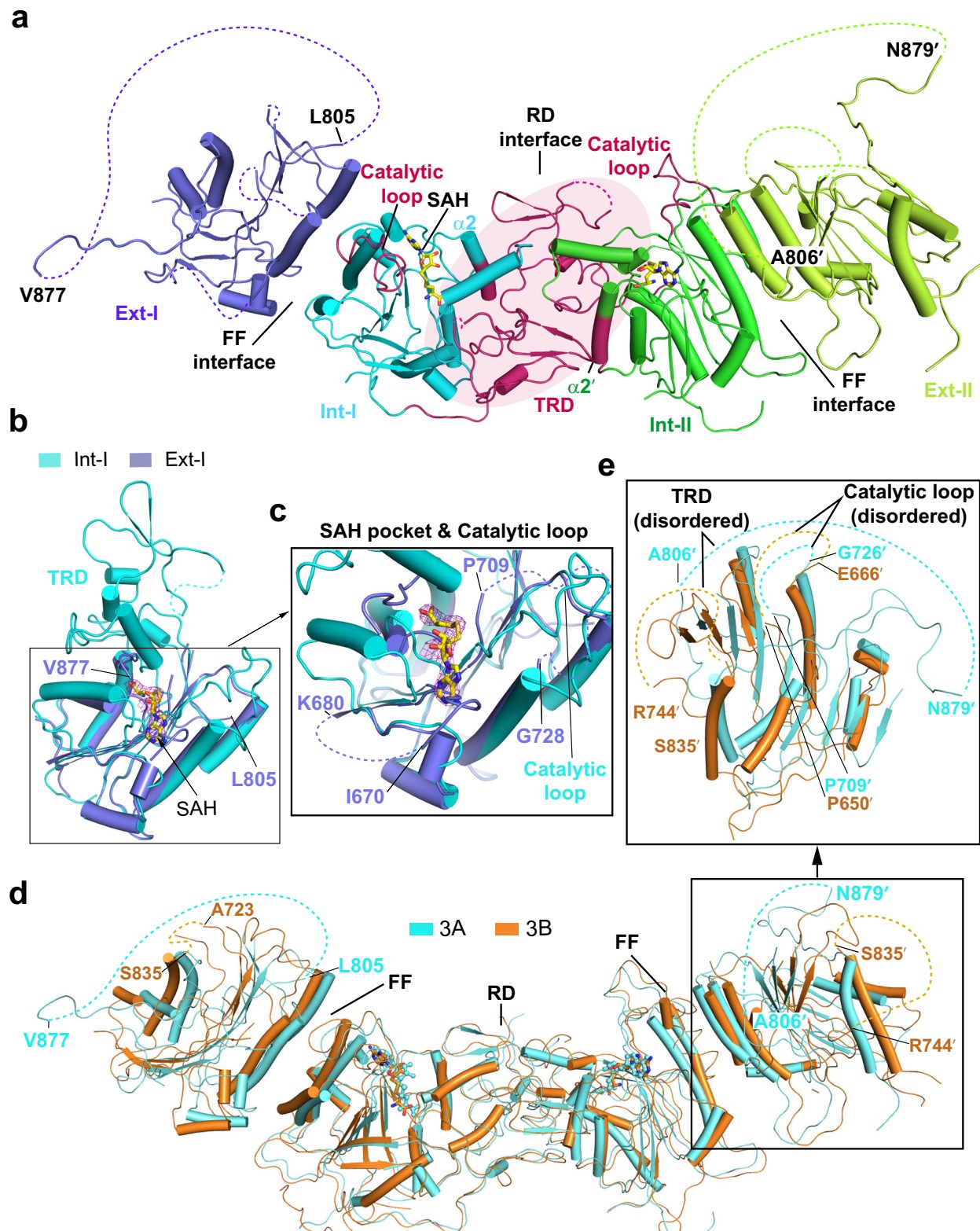

**Fig. 1 | Crystal structure of homotetrameric DNMT3A MTase domain. a** Ribbon diagram of the DNMT3A homotetramer, with individual subunits Ext-I, Int-I, Int-II, and Ext-II labeled and color-coded. The RD and FF interfaces are marked by solid lines. Regions that are disordered, including the TRD (labeled with the delimiting residues) in the two external subunits are indicated by black dashed lines. The corresponding regions in the central subunits are colored in warm pink. **b** Structural overlay of the Ext-I (slate) and Int-I (aquamarine) subunits. The Fo−Fc omit map (contoured at 2.0 σ) of the SAH molecule is shown as a purple mesh. **c** Close-up view of the overlaid SAH-binding pocket and catalytic loop between Ext-I and Int-I subunits, indicating that these regions are disordered in the Ext-I subunit but ordered in the Int-I subunit. **d** Structural overlay of DNMT3A homotetramer (aquamarine) and DNMT3B homotetramer (orange) (PDB 8EIH). Both protein complexes show well-defined central subunits but partially disordered external subunits. **e** Close-up view of the overlaid external subunits between DNMT3A and DNMT3B, indicating the structural disorder of the regions underpinning the RD interface.

(Fig. 1d, e)[36]. Together, these observations suggest oligomerization-coupled RD interface folding as a recurrent mechanism for the DNMT3 family of DNA methyltransferases.

It is worth noting that the regions that remain structurally ordered in the external subunits of DNMT3A align well with the C-terminal domain of DNMT3L, with root-mean-square deviation (RMSD) of 1.3 Å for 104 aligned Cα atoms (Supplementary Fig. 1d). On the other hand, the structurally disordered regions of the external subunit of DNMT3A coincide with the sequences that are missing in DNMT3L, reflecting an evolutionary link between DNMT3A and DNMT3L (Supplementary Figs. 1d and 2).

## RD interface variation underpins distinct oligomerization behavior between DNMT3A and DNMT3B

Structural comparison of homotetrameric DNMT3A and DNMT3B reveals a nearly identical RD interface (Fig. 2a), in line with their high sequence identity (Supplementary Fig. 2). Of note, residue R882 of DNMT3A homotetramer and the corresponding DNMT3B R823 are both solvent-exposed and peripheral to the RD interface (Fig. 2a). On the other hand, it is also evident that subtle sequence variations between DNMT3A and DNMT3B give rise to distinct inter-molecular interactions in this region. For instance, in DNMT3A homotetramer, residue R676 from one central subunit engages in electrostatic

attraction with residue E820′ from the other central subunit; in DNMT3B homotetramer, a arginine-to-lysine replacement at the corresponding site (K617) increased the contact distance by ~2 Å (Fig. 2a–c). In addition, a methionine-to-threonine replacement also led DNMT3A M674 to alter intermolecular contacts at the RD interface from its counterpart (T615) in DNMT3B. In accordance with these aminoacid variations, the formation of the RD interface in DNMT3A leads to a slightly larger buried surface area than that in DNMT3B (925 Å² in DNMT3A vs 864 Å² in DNMT3B).

Next, we asked whether the RD interface variations differentiate the oligomerization behaviors of DNMT3A and DNTM3B. To address this, we performed size-exclusion chromatography analysis on the C-terminal fragments of DNMT3A and DNMT3B, each harboring the PWWP, ADD and MTase domains, which appear biochemically more amenable than full-length proteins. WT DNMT3A and DNMT3B fragments eluted in a volume largely corresponding to their tetrameric forms, consistent with previous observations that both proteins are dominated by tetrameric forms in solution (Fig. 2d, e)[33–35,40,42]. On the other hand, the DNMT3A R882H mutation shifts the DNMT3A elution volume to that corresponding to increased oligomerization (Fig. 2d), as previously reported[40,42]. In contrast, the DNMT3A R882H-corresponding mutation in DNMT3B, R823H, failed to affect the elution volume of DNMT3B appreciably (Fig. 2e). This observation,

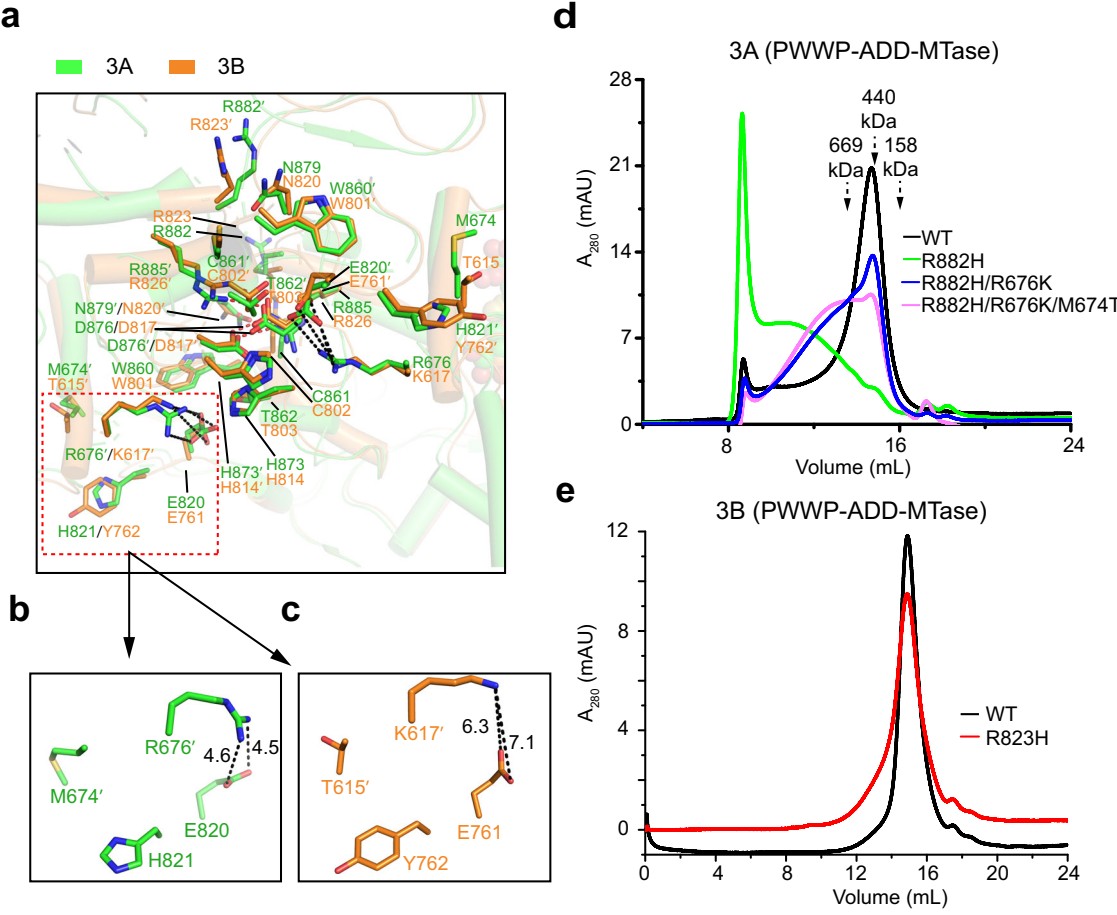

**Fig. 2 | RD interface variation underpins distinct oligomerization behavior between DNMT3A and DNMT3B. a** Structural overlay of the RD interfaces of DNMT3A and DNMT3B (PDB 8EIH), with residues involved in intermolecular interactions shown in stick representation. Hydrogen-bonding and electrostatic interactions are shown as red and black dashed lines, respectively. **b** Close-up view of DNMT3A-unique RD interface residues M674, R676, and H821, with the inter-molecular sidechain distances between R676′ (prime symbol denotes residue from the symmetry-related subunit) and E820 labeled in unit of Å. **c** Close-up view of

DNMT3B-unique RD interface residues T615, K617, and Y762, with the side-chain distances between K617′ and E761 from the symmetry-related subunit labeled in unit of Å. **d** Size-exclusion chromatography analysis of DNMT3A fragment comprised of the PWWP, ADD and MTase domains, WT, R882H, R882H/R676K, or R882H/R676K/M674T mutant. The elution volumes for molecular weight standards (thyroglobulin: 669 kDa, ferritin: 440 kDa, adolase: 158 kDa) are indicated by arrows. **e** Size-exclusion chromatography analysis of DNMT3B fragment comprised of the PWWP, ADD and MTase domains, WT or R823H mutant.

consistent with a previous study that the mouse DNMT3B R829H mutant (equivalent to R823H in human DNMT3B) shows a comparable DNA methylation activity as that of WT DNMT3B[51], suggests a minimal impact of this mutation on DNMT3B oligomerization. We then introduced R676K single mutation or M674T/R676K double mutation to R882H-mutated DNMT3A (DNMT3A^R882H) to mimic the corresponding site in DNMT3B. We observed that these DNMT3B-converting mutations led to a shift of the elution volume of DNMT3A^R882H toward the tetrameric state, indicating a polymerization-attenuating effect (Fig. 2d).

It is worth noting that the R882C mutation, which constitutes the second most frequent mutation next to R882H in AML and causes a DNA hypomethylation effect in cells and in vivo[40], exhibits a polymerization-promoting effect similar to that of the R882H mutation (Supplementary Fig. 3a). On the other hand, introducing the DNMT3B-converting mutations to R882C-mutated DNMT3A (DNMT3A^R882C) also shifts the size-exclusion chromatography profile toward that corresponding to a tetrameric assembly (Supplementary Fig. 3a), suggesting that the polymerization-attenuating effect of the DNMT3B-converting mutations also applies to DNMT3A^R882C. Together, these observations suggest that the sequence variations at the RD interface of DNMT3A and DNMT3B give rise to their differential susceptibility in oligomerization to interface mutations.

### Structure basis for the DNMT3A R882 mutations

Despite numerous efforts, we were unable to obtain crystals of DNMT3A^R882H or DNMT3A^R882C with sufficient quality for structure determination, due to their strong polymerization behavior and low solubility in solution. In light of the fact that introducing a secondary DNMT3B-converting mutation, DNMT3A R676K, partially reduces the polymerization of DNMT3A^R882H/C (Fig. 2d and Supplementary Fig. 3a), we set out to prepare R882H/R676K- and R882C/R676K-mutated DNMT3A MTase domains for crystallization, which led to successful structure determination of DNMT3A^R882H/R676K at 2.48 Å resolution and DNMT3A^R882C/R676K at 3.2 Å resolution (Fig. 3 and Supplementary Table 1).

The structures of DNMT3A^R882H/R676K and DNMT3A^R882C/R676K both reveal a tetrameric assembly similar to that of WT DNMT3A, with an RMSD of 0.29 Å and 0.33 Å over 800 and 811 aligned Cα atoms, respectively (Fig. 3a–e). As with WT DNMT3A, the central subunits are well defined, each containing a largely intact TRD and SAH-binding pocket, whereas the corresponding regions are disrupted in the two external subunits (Fig. 3a, d and Supplementary Fig. 3b, c). Note that the introduction of the R676K mutation does not lead to an appreciable structural change for its surrounding region (Fig. 3b), except for an increased distance between site 676 and residue E820′ from the symmetry-related subunit (Supplementary Fig. 3d), explaining why the R676K mutation partially reverses the polymerization behavior of DNMT3A^R882H and DNMT3A^R882C (Fig. 2d and Supplementary Fig. 3a). Consistent with these observations, introducing the R676K or M674T/R676K mutation to maltose-binding protein (MBP)-tagged DNMT3A MTase domain shifted its elution volume on a size-exclusion chromatography column closer to that corresponding to the homotetrameric form, and slightly increased its DNA methylation activity in vitro (Supplementary Fig. 3e, f).

Detailed analysis of DNMT3A^R882H/676K and DNMT3A^R882C/R676K reveals structural traits at the RD interface distinct from that of WT DNMT3A. First, unlike R882 in WT DNMT3A which is conformationally flexible with poor electron density, residues H882 in DNMT3A^R882H/R676K and C882 in DNMT3A^R882C/R676K are associated with well-defined electron density, suggesting reduced sidechain flexibility (Supplementary Fig. 3g–i). Furthermore, unlike residue R882 in WT DNMT3A that makes no appreciable inter-molecular contacts at the RD interface, both H882 and C882 engage in notable intermolecular interactions in their respective complexes (Fig. 3c, e and Supplementary Fig. 3j).

For DNMT3A^R882H/R676K, residue H882 is involved in a sidechain stacking interaction with residue N879′ from the symmetry-related subunit (Fig. 3c), reminiscent of what was previously observed for residue H882 in the DNMT3A^R882H–DNMT3L–DNA complex (Supplementary Fig. 4a). In addition, residue H882 makes van der Waals contacts with residues L883 and Q886 from the same DNMT3A subunit, which may further stabilize the intermolecular H882–N879′ interaction (Fig. 3c). For DNMT3A^R882C/R676K, the sidechain sulfhydryl group of residue C882 forms a reciprocal pair of weak hydrogen-bonding interactions with the backbone carbonyl group of residue M880′ from the symmetry-related subunit (Fig. 3e). In addition, the side chain of residue C882 engages in van der Waals contacts with residue C861 from the same subunit and residue N879′ from the symmetry-related subunit (Fig. 3e). These observations suggest that mutation of R882 into a histidine or cysteine leads to an enhanced intermolecular interaction at the RD interface, which consequently promotes DNMT3A polymerization.

To examine the link between the intermolecular interactions described above and the polymerization-promoting effect of the R882 mutations, we further generated the N879A/R882H-mutated DNMT3A (DNMT3A^R882H/N879A) protein. Size-exclusion chromatography analysis of DNMT3A^R882H/N879A reveals an elution volume corresponding to a tetrameric assembly (Supplementary Fig. 4b), suggesting that impairment of the H882–N879′ intermolecular interaction indeed reversed the polymerization-promoting effect of DNMT3A R882H mutation. We then solved the crystal structure of DNMT3A^R882H/N879A at 2.65 Å resolution (Supplementary Fig. 4c–f and Supplementary Table 1). The structure of DNMT3A^N879A/R882H reveals that whereas residue H882 assumes a well-defined conformation to interact with residues L883 and Q886 in the same molecule (Supplementary Fig. 4d, e), its intermolecular contact was reduced by the N879A mutation (Supplementary Fig. 4d). These observations reinforce the notion that the newly introduced intermolecular contacts by the R882 mutations give rise to the enhanced oligomerization behavior of DNMT3A.

### Biochemical rescue of DNMT3A^R882H and DNMT3A^R882C by DNMT3B-converting mutations

The observation that the DNMT3B-converting mutations may attenuate the polymerization-promoting effect of DNMT3A^R882H/DNMT3A^R882C raises the possibility of rescuing the phenotype of DNMT3A^R882H or DNMT3A^R882C via such DNMT3B-converting mutations. To test this notion, we set out to evaluate the effect of DNMT3B-converting mutations on the biochemical and enzymatic properties of DNMT3A^R882H and DNMT3A^R882C, focusing on the MTase domain that has similar kinetic parameters as full-length enzyme[44]. In addition, we adopt a previous strategy that resorts to a fused MBP tag to overcome the low solubility of DNMT3A^R882H and DNMT3A^R882C[43]. First, DNMT3A MTase eluted in a much broader peak on a size-exclusion chromatography column than that of the heterotetrameric complex between DNMT3A MTase and the DNMT3L C-terminal domain (Supplementary Fig. 5a, b), suggesting that DNMT3A, like DNMT3B[36], may exist as a mixture of homotetramer and other alternative assembly states in solution. Consistently, our measurement of the hydrodynamic radii ($R_h$) of DNMT3A and DNMT3B using the dynamic light scattering (DLS) method reveals that, under the experimental condition, WT DNMT3A shows a size distribution profile centered at $R_h$ of 22 nm, similar to that of DNMT3B (Fig. 4a and Supplementary Fig. 5c). Notably, the DNMT3A R882H and R882C mutations each shift DNMT3A to a higher molecular weight population, centered at $R_h$ of 40 nm and 60 nm, respectively (Fig. 4a), reinforcing the notion that these mutations promote DNMT3A polymerization. On the other hand, introducing the DNMT3B-converting mutation, either R676K or M674T/R676K, led to a size distribution similar to that for WT DNMT3A (Fig. 4a). It is worth noting that our thermal shift assays indicated that these DNMT3B-converting mutations do not affect the stability of DNMT3A, DNMT3A^R882H, or DNMT3A^R882C appreciably (Supplementary Fig. 5d–g).

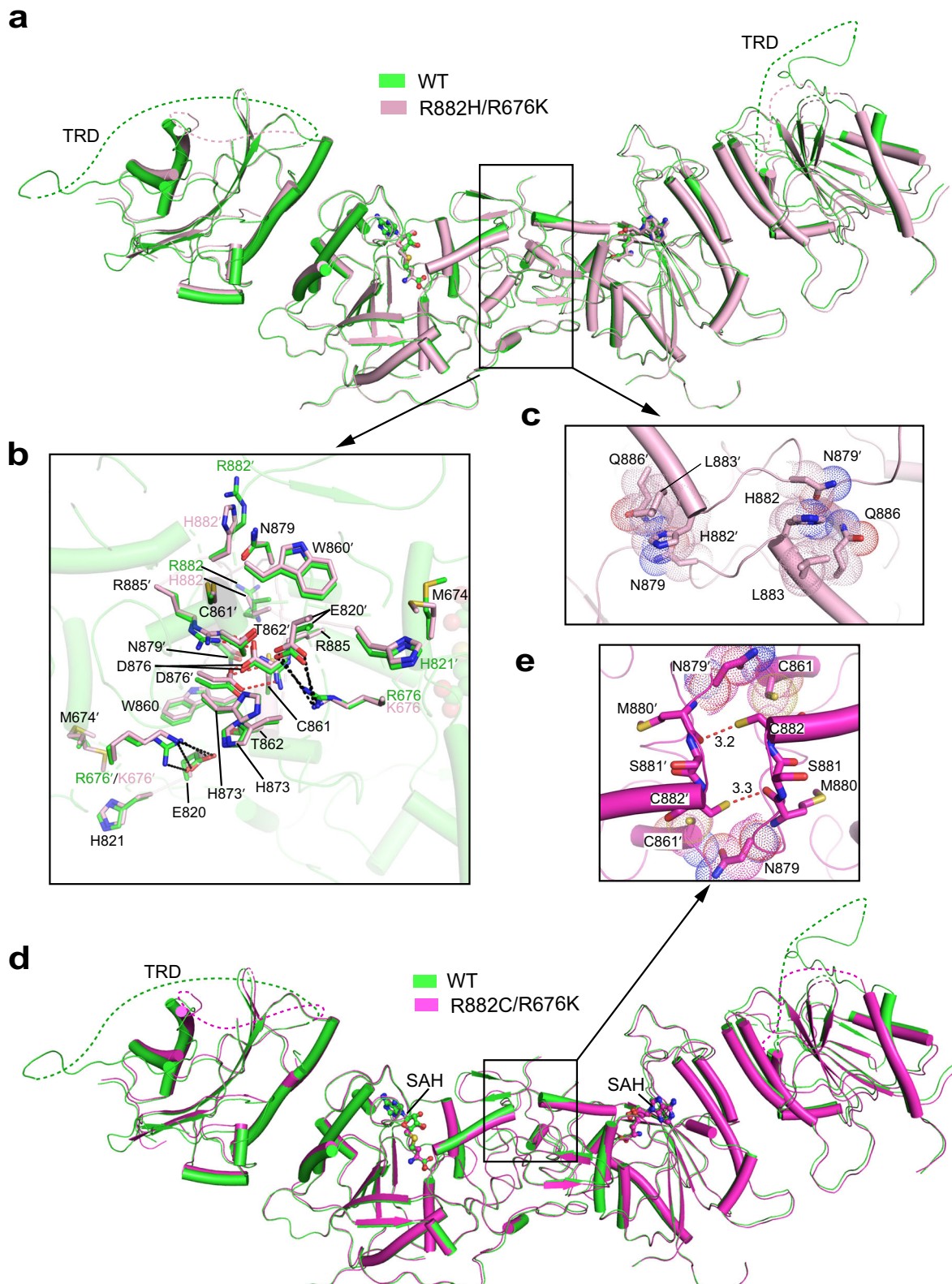

**Fig. 3 | Structural characterization of R882-mutated DNMT3A. a** Structural overlay between WT DNMT3A (green) and DNMT3A^R882H/R676K (pink). The disordered TRDs in the external subunits are shown as dashed lines. **b** Structural comparison of the RD interface between WT DNMT3A and DNMT3A^R882H/R676K, with residues involved in intermolecular interactions shown in stick representation. Hydrogen bonds are shown as dashed lines. **c** Close-up view of residues involved in H882-mediated interactions in DNMT3A^R882H/R676K. The van der Waals radii of their sidechain atoms are depicted by dot representation. **d** Structural overlay between WT DNMT3A (green) and DNMT3A^R882C/R676K (magenta). The disordered TRDs are shown as dashed lines. **e** Close-up view of residues involved in C882-mediated contacts in DNMT3A^R882C/R676K. The van der Waals radii of their sidechain atoms are depicted by dot representation. Hydrogen bonds are shown as dashed lines, with distances labeled in unit of Å.

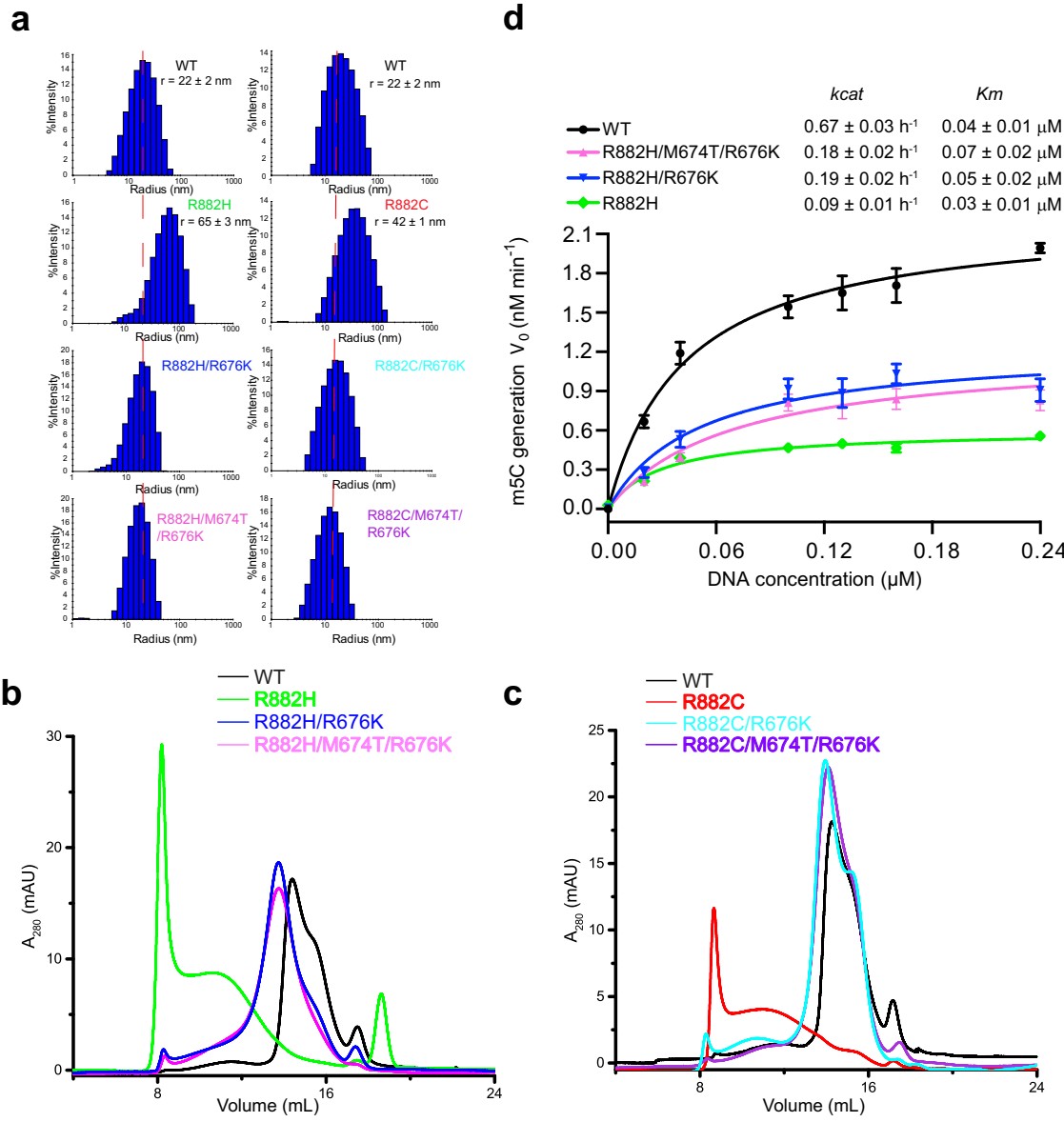

**Fig. 4 | Biochemical characterization of DNMT3B-converting mutations. a** DLS plot of MBP-tagged DNMT3A MTase domain, WT, and mutant. **b, c** Size-exclusion chromatography analysis of MBP-tagged DNMT3A MTase domain examining the mutational effect of DNMT3A$^{R882H}$ (**b**) or DNMT3A$^{R882C}$ (**c**) in the absence or presence of additional DNMT3B-converting mutations. **d** Steady-state enzymatic kinetics of WT, R882H-, R882H/R676K-, and R882H/M674T/R676K-mutated MBP-tagged DNMT3A MTase domain. Data are mean ± SD ($n = 3$, biological replicates). Source data are provided as a Source Data file.

These observations, consistent with size-exclusion chromatography analysis of DNMT3A MTase carrying the corresponding RD interface mutations (Fig. 4b, c), support the role of the DNMT3B-converting mutations in attenuating DNMT3A R882H/R882C-promoted polymerization.

To examine how the RD interface mutations affect the DNA methylation activities of DNMT3A, we further performed in vitro DNA methylation analysis on a CpG-containing DNA duplex at various DNMT3A protein concentrations. As previously observed, the activity increase over increasing protein concentrations of DNMT3A$^{R882H}$, due to the formation of polymers, is substantially slower than that of WT DNMT3A (Supplementary Fig. 5h)[32,40,42,43]. However, the slope for such a concentration-dependent activity is partially restored for DNMT3A$^{R882H/R676K}$ and DNMT3A$^{R882H/M674T/R676K}$ (Supplementary Fig. 5h), consistent with the polymerization-attenuating effect of the R676K and M674T/R676K mutations. It is noted that DNMT3A$^{R882C}$, DNMT3A$^{R882C/R676K}$, and DNMT3A$^{R882C/M674T/R676K}$ all show severely

reduced DNA methylation activity (Supplementary Fig. 5i), suggesting that whereas the DNMT3B-converting mutations attenuate DNMT3A$^{R882C}$ polymerization, the activity loss of DNMT3A$^{R882C}$ is rather dominated by a defect in substrate binding. Next, we performed steady-state kinetic assays to dissect the impact of DNMT3B-converting mutations on the enzymatic kinetics of DNMT3A$^{R882H}$ (Fig. 4b and Supplementary Fig. 6a–d). A $k_{cat}$ of 0.67 h$^{-1}$ was determined for WT DNMT3A, whereas a ~7-fold lower $k_{cat}$ was determined for DNMT3A$^{R882H}$, consistent with a previous observation[44]. However, introducing the DNMT3B-converting mutations, R676K or R676K/M674T, to DNMT3A$^{R882H}$ led to a ~2-fold increase of $k_{cat}$, reflecting an effect of these mutations on restoring the tetramer concentration. The fact that the DNA methylation activities of DNMT3A$^{R882H/R676K}$ and DNMT3A$^{R882H/M674T/R676K}$ remain lower than that of WT DNMT3A is in line with the fact that the R882H mutation, in addition to the polymerization-promoting effect, impaired the DNA-binding activity of DNMT3A at the RD interface (Supplementary Fig. 6e–h), as observed

previously[32]. Together, these results further support that the DNMT3B-converting mutations attenuate DNMT3A[R882H] and DNMT3A[R882C] polymerization, which consequently leads to an enrichment of DNMT3A tetramers, the most active enzyme form in vitro[34,40].

## Decreasing DNMT3A binding onto chromatin by disease-related hotspot mutation

To assess how exactly the enhanced oligomerization of DNMT3A hotspot mutant may affect DNMT3A's function such as chromatin targeting, we turned to the TF-1 leukemia cell model[39]. Here, we ectopically expressed in TF-1 cells either WT or R882-mutated DNMT3A (R882C or R882H) or the one carrying a DNMT3B-converting mutation (namely, R676K or M674T/R676K alone, or its combination with a R882 hotspot mutation), with empty vector used as mock control (Supplementary Fig. 7a). Using cleavage under targets & tagmentation (CUT&Tag)[52], we mapped genome-wide binding of Myc-tagged DNMT3A, either WT or mutant, in TF-1 cells, with murine NIH3T3 fibroblasts expressing Myc-tagged WT DNMT3A added as a spike-in control for normalization. Considering that DNMT3A binding reportedly correlates with H3K36me2, a histone mark bound directly by the PWWP domain of DNMT3A[53], we additionally conducted H3K36me2 CUT&Tag. The called peaks of DNMT3A, WT or mutant, are highly correlated with each other, suggesting similarity in their peak locations, and there is an expected positive correlation between DNMT3A and H3K36me2, a pattern not affected by the mutation (Supplementary Fig. 7b). Genomic regions enrichment of annotations tool (GREAT) analysis of DNMT3A peaks in TF-1 cells revealed the enrichment of pathways associated with erythrogenesis and immunity-related processes (Supplementary Fig. 7c). Genomic annotation of the DNMT3A peaks also showed that WT and all mutant DNMT3A largely had similar distribution among the promoter, intron, intergenic and repetitive regions (results from two independent experiments shown in Fig. 5a and Supplementary Fig. 7d). Next, we more closely analyzed the DNMT3A binding density by using reads per kilobase of transcript per million reads (RPKM) of the called peaks followed by normalization with spike-in controls. Compared with WT, both hotspot mutations (R882C and R882H) significantly reduced DNMT3A binding at all annotated genomic regions, regardless of the promoter, exon, intron, intergenic regions, or repetitive elements (results from two independent experiments shown in Fig. 5b, c and Supplementary Fig. 7e, f; R882C or R882H vs WT), as exemplified by what was seen at LINE-1 repetitive elements and the LMO2 gene (Fig. 5d). Meanwhile, the introduction of an additional DNMT3B-converting mutation significantly rescued the hotspot mutant-associated binding defects at all regions (Fig. 5b–d and Supplementary Fig. 7e, f; see R882C/R676K vs R882C, or R882H/M674T/R676K vs R882H in two independent experiments). Such a reversing effect by R676K was not seen in the context of WT DNMT3A, suggesting that effect is indeed specific to hotspot mutation—as a matter of fact, R676K DNMT3A alone exhibited a rather mild change in overall binding and even a somewhat decreased binding when compared to WT control (Fig. 5b–d and Supplementary Fig. 7e, f; R676K vs WT in two independent experiments). Taken together, the DNMT3A hotspot mutant exhibits an overall decreased chromatin binding, which can be attributed at least partly to polymerization induced by hotspot mutation.

## Suppression of DNMT3A hotspot mutation-induced CpG hypomethylation and cytokine-independent growth in TF-1 cells

Previous studies have shown that the DNMT3A hotspot mutations do not lead to a substantial change in global DNA methylation; rather, they cause focal hypomethylation at a subset of CpG sites[40,41]. Using mass spectrometry-based quantification, we measured the global 5-methyl-2′-deoxycytidine (5-mdC) levels in the genomic DNA of TF-1 cells with expression of the above-mentioned DNMT3A. We found that, except the two DNMT3B-converting mutants in the context of WT

DNMT3A (R676K and M674T/R676K), the ectopically expressed DNMT3A had no significant effect on global DNA methylation when compared to mock (Supplementary Fig. 8a). The two DNMT3A hotspot mutants (R882H or R882C alone) appeared to decrease global methylation but changes were not statistically significant in our experiments (Supplementary Fig. 8a), consistent with previous findings[40,41]. To further assess site-specific DNA methylation, we next employed the Infinium methylationEPIC array, which covers a total of 865,917 representative CpG sites in the human methylome. Principal component analysis (PCA) of array profiles showed that TF-1 cells transduced with a DNMT3A hotspot mutant, either R882C or R882H, are more closely related to one another than other cell groups (Fig. 6a). The DNMT3B-converting mutation (R676K or M674T/R676K) significantly shifted the overall methylome patterns towards those of mock controls (Fig. 6a). In contrast, the same polymerization-attenuating mutations did not have such an effect in the context of WT DNMT3A (Fig. 6a; R676K or M674T/R676K vs WT). Furthermore, we defined differentially methylated CpGs (DMCs). Consistent with our previous report[39], both R882C and R882H mainly induced hypomethylation when compared to mock-treated, with DMCs exhibiting significant hypomethylation being 22–25-fold more than those exhibiting hypermethylation (Supplementary Fig. 8b, c; Δbeta value greater than 10% and q less than 0.01 in paired t test). There is overlap at a majority of R882C- and R882H-associated DMCs, either hypomethylated or hyper-methylated (Supplementary Fig. 8d, e). In stark contrast, the effect induced by WT DNMT3A was predominantly hypermethylation (Supplementary Fig. 8f). Also, compared to the single R676K mutation, the double converting mutation M674T/R676K had more consistent methylome-shifting effects in the context of both R882C and R882H; meanwhile, the effects by two polymerization-attenuating mutations (R676K and M674T/R676K) on R882H were generally less than those on R882C (Supplementary Fig. 8g, h, i). No significant DNA methylation rescue effect by R882H/R676K, when compared to R882H, was observed (data not shown). We thus focused on R882C/R676K, R882C/M674T/R676K, and R882H/M674T/R676K. Compared to the DNMT3A R882C controls, the introduction of an additional DNMT3B-converting mutation (R676K or M674T/R676K) induced re-methylation at a significant portion of the R882C-induced hypomethylated DMCs (Fig. 6b, c and Supplementary Fig. 8g, h), as exemplified by those associated with EFL4, FOSL2, KDM2A, and HOXB3 (Fig. 6d). A significant effect by M674T/R676K on converting the R882H-associated hypomethylation was also observed (Fig. 6c, d), albeit with a smaller effect than what was seen in the context of DNMT3A R882C (Supplementary Fig. 8i vs Supplementary Fig. 8h). A total of 7,196 CpGs exhibiting hypomethylation upon R882C transduction were significantly re-methylated by both converting mutations (Fig. 6b; R882C/R676K and R882C/M674T/R676K vs R882C), and genes associated with these CpGs are enriched in the signatures related to hematopoietic function and pathogenesis, confirming a role of DNMT3A in normal hematopoiesis (Supplementary Fig. 8j).

Aside from a predominant role in inducing CpG hypomethylation, previous studies also showed an off-target effect by DNMT3A hotspot mutation[32,41,45,46]. Of note, DNMT3A[R882H] exhibits a DNMT3B-like enzymatic specificity, with a marked preference for guanine at the +1-flanking site[45,46]. Ectopic expression of DNMT3A[R882H] in mouse embryonic stem cells with DNMT3A/DNMT3B double KO substantially rescued DNA methylation at the target sites of DNMT3B, but not DNMT3A[46]. Consistent with these studies, the DNMT3A hotspot mutant-associated hyper-methylated CpGs, as defined in our array profiles (765 sites; Supplementary Fig. 8e), favor purine (guanine or G) and pyrimidine (cytosine or C) at the +1 and −1 flanking positions, respectively (Supplementary Fig. 8k), when using all 865,917 representative CpG sites as background. Meanwhile, the hotspot mutant-associated hypomethylated CpGs (20,888 sites; Supplementary Fig. 8e) showed a

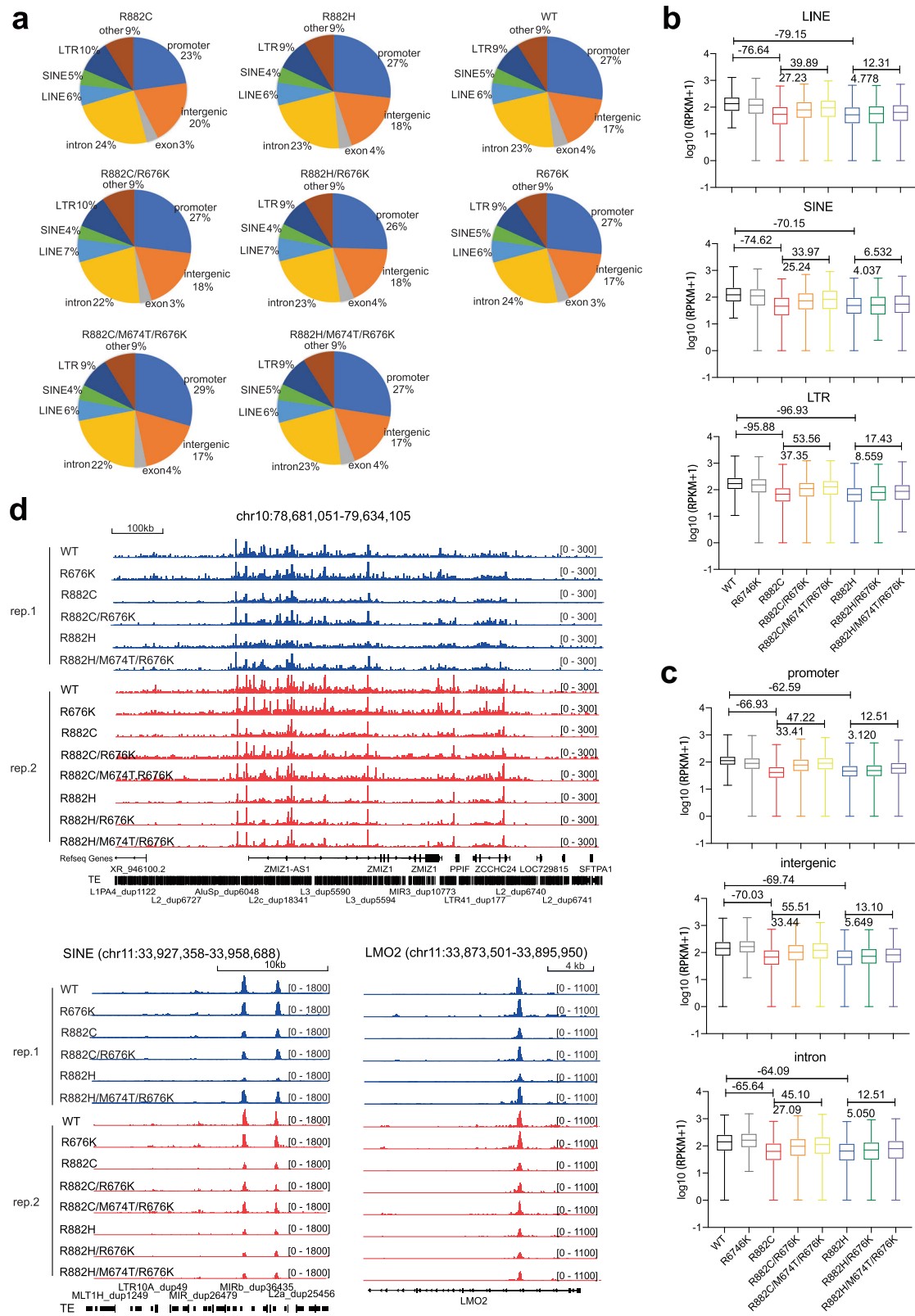

preference towards pyrimidine (thymine or T) and purine (adenine or A) at the +1 and −1 flanking positions, respectively (Supplementary Fig. 8l), which is a reported feature of WT DNMT3A enzyme[15,54]. Again, when compared to WT, the DNMT3A hotspot mutant exhibited a decreased overall binding at those 20,888 DMCs exhibiting hypomethylation upon transduction of the hotspot mutation, and such a defect was partially rescued by the DNMT3B-converting mutation (Supplementary Fig. 8m).

Our above data show that DNMT3A hotspot mutant exhibits a partial loss of WT DNMT3A-related activities and has an off-target effect; however, the relative contribution of the two effects to leukemogenic transformation remains unclear. As the DNMT3B-converting mutation predominantly induces re-methylation, we reasoned that such a converting mutation represents a strategy for dissecting the specific contribution of the loss-of-function effect associated with the

**Fig. 5 | Rescue DNMT3A hotspot mutation-induced decrease of chromatin binding. a** Distribution of the indicated WT or mutant DNMT3A CUT&Tag peaks (anti-Myc antibody used in CUT&Tag for Myc-tagged DNMT3A) among various genomic regions in TF-1 cells. **b, c** The averaged intensity of DNMT3A binding, as assayed by CUT&Tag and spike-in control-based normalization, at various repetitive element classes (**b** long terminal repeat (LTR), long interspersed nuclear elements (LINEs), or short interspersed nuclear elements (SINEs)), or at the genic (promoter, exon and intron) and intergenic regions (**c**), in TF-1 cells transduced with the indicated DNMT3A, WT or mutant (x-axis). The y-axis shows the log10 value of Reads per kilobase per million reads mapped (RPKM). The whiskers are minimum to maximum, the box depicts the 25th–75th percentiles, and the line in the middle of the box is plotted at the median. LINE, $n = 445$; SINE, $n = 316$; LTR, $n = 644$; promoter, $n = 1947$; intergenic, $n = 1184$; and intron, $n = 1659$. The two-tailed Wilcoxon test was used. Numbers on the top of the columns indicate the difference in the median values of RPKM between the two compared groups. **d** Integrative genomics viewer (IGV) views for spike-in-control-normalized binding of the indicated DNMT3A at a repetitive element-rich region in chromosome 10, a SINE element in chromosome 11, and LMO2 in TF-1 cells. Rep 1 and 2, two independently performed experiments.

DNMT3A hotspot mutant. Towards this end, we scored for cytokine-independent growth of TF-1 cells, which was known to be associated with the DNMT3A hotspot mutant and not WT DNMT3A[39] (Fig. 6e–g; vector vs WT, R882C or R882H). Here, we found that the DNMT3B-converting mutation alone (R676K and M674T/R676K) had no effect on TF1 cells (Fig. 6e). But in the context of DNMT3A R882C or R882H transduction, introducing an additional M674T/R676K or R676K mutation had significant effect on reversing the cell growth advantage conferred by the hotspot mutant, with effect of M674T/R676K significantly stronger than that of R676K (Fig. 6f, g). Based on these observations, we conclude that enhanced oligomerization of the DNMT3A hotspot mutant underlies the mutation-induced DNA hypomethylation and cytokine-independent growth in TF-1 cells. Whereas the off-target effect does exit, attenuating the polymerization by itself is sufficient to suppress the dominant-negative effect of DNMT3A hotspot mutations.

## Discussion

How dysregulation of DNA methyltransferases leads to human diseases remains a fundamental question. DNMT3A is not only essential for establishing de novo DNA methylation in germ cells and early embryogenesis, but also plays a critical role in maintaining the dynamic DNA methylation landscape in differentiated cells. How DNMT3A mutations contribute to the pathogenies of AML and other diseases is not fully understood. This study focuses on a structural understanding of DNMT3A oligomerization and how DNMT3A R882H and R882C, which are prevalent in AML and associated with poor prognosis, lead to aberrant DNMT3A oligomerization. Through a combined structural and functional analysis, our study reveals the molecular basis for the pathogenic effect of these mutations and identifies polymerization attenuation as a strategy of therapeutic intervention, with important implications in drug development against AML and other DNMT3A-associated diseases.

### Structure and mechanism of DNMT3A/DNMT3B homo-oligomerization

This study provides mechanistic insights into the assembly of DNMT3A and its disease mutations R882H and R882C. The two central subunits of DNMT3A homotetramer form an integrated substrate-binding platform resembling that of the DNMT3A homodimer within the DNMT3A–DNMT3L complex, whereas the two external subunits abrogate their substrate-binding sites due to the lack of a folded TRD and cofactor-binding pocket. This conformational asymmetry between the central and external subunits, coupled with the integrity of the RD interface, was similarly observed for the cryo-EM structure of DNMT3B, suggesting a common assembly mechanism within the DNMT3 family. The coupling between the RD interface formation and folding of the substrate-binding sites is thermodynamically governed by compensated enthalpy and entropy, which influences the equilibrium of DNMT3A/DNMT3B oligomerization. Along the line, our cryo-EM study of DNMT3B reveals the co-existence of homotetramer with other alternative assemblies, such as homohexamer and homotrimer, in solution[36]. This study, through structural and biochemical characterization of DNMT3A homo-oligomer, suggests that the assembly

of DNMT3A may undergo a similar transition between alternative oligomerization states in solution (Fig. 7). Consistently, it has been reported that DNMT3A cooperatively binds and methylates DNA substrates[31,34,51,55,56], which is presumably accompanied by the oligomerization-coupled structural ordering of the RD interface, thereby permitting efficient DNA methylation in a DNA-dense environment. On the other hand, in the absence of DNA, the R882H mutation-induced excessive oligomerization (i.e., polymerization) of DNMT3A causes a decrease of the concentration of DNMT3A particles in solution, leading to a reduced DNA methylation activity in vitro and in cells[40,42]. In this context, maintaining a balanced equilibrium of the assembly states of DNMT3A and DNMT3B appears crucial to their functionalities, which merits additional studies in the future.

### Structural basis for R882H- and R882C-promoted protein polymerization

Our previous studies have demonstrated that DNMT3A R882 engages in substrate binding via direct DNA contact, as well as an intramolecular interaction that modulates the CpG recognition by the TRD loop[15,27,32]. Here, our comparative structural analyses of DNMT3A, DNMT3A^R882H/R676K, and DNMT3A^R882C/R676K reveal that the AML-associated R882H and R882C mutations do not impact the DNMT3A conformation substantially; rather, they introduce a subtle change to the chemical environment of the RD interface. Unlike WT DNMT3A in which residue R882 extends its side chain toward solvent and makes no appreciable intermolecular interactions, R882H- and R882C-mutated DNMT3A proteins each confer site 882 with a smaller amino acid and reduced conformational flexibility. Furthermore, the R882H and R882C mutations both bring additional intermolecular interactions to the RD interface: residue H882 in DNMT3A^R882H/R676K engages in a sidechain packing interaction with residue N879′ from the symmetry-related subunit, while residue C882 in DNMT3A^R882C/R676K forms a sidechain hydrogen bond with the backbone carbonyl of residue M880′ from the symmetry-related subunit. These newly introduced intermolecular interactions conceivably strengthen the oligomeric association between DNMT3A molecules via the RD interface, thereby promoting the polymerization behaviors of DNMT3A (Fig. 7).

### Fine-tuning of protein oligomerization as a strategy for therapeutic intervention

How DNMT3A and DNMT3B have evolved into two functionally overlapped but distinct DNA methyltransferases remains an unresolved question. Our previous studies demonstrated that the sequence variations at the catalytic loop and RD interface of DNMT3A and DNMT3B underpin their distinct CpG-recognition mechanisms[15]. In this study, we further uncovered the distinct oligomerization properties between DNMT3A and DNMT3B, evidenced by their differential response to the DNMT3A R882H-corresponding mutation. Introducing the DNMT3B-converting mutation, DNMT3A R676K or M674T/R676K, to R882H-mutated DNMT3A shifts its oligomerization from a polymeric state toward a tetrameric state, suggesting that such a distinct oligomerization behavior is at least in part attributed to the sequence variation within the RD interface. In comparison with R882H- and

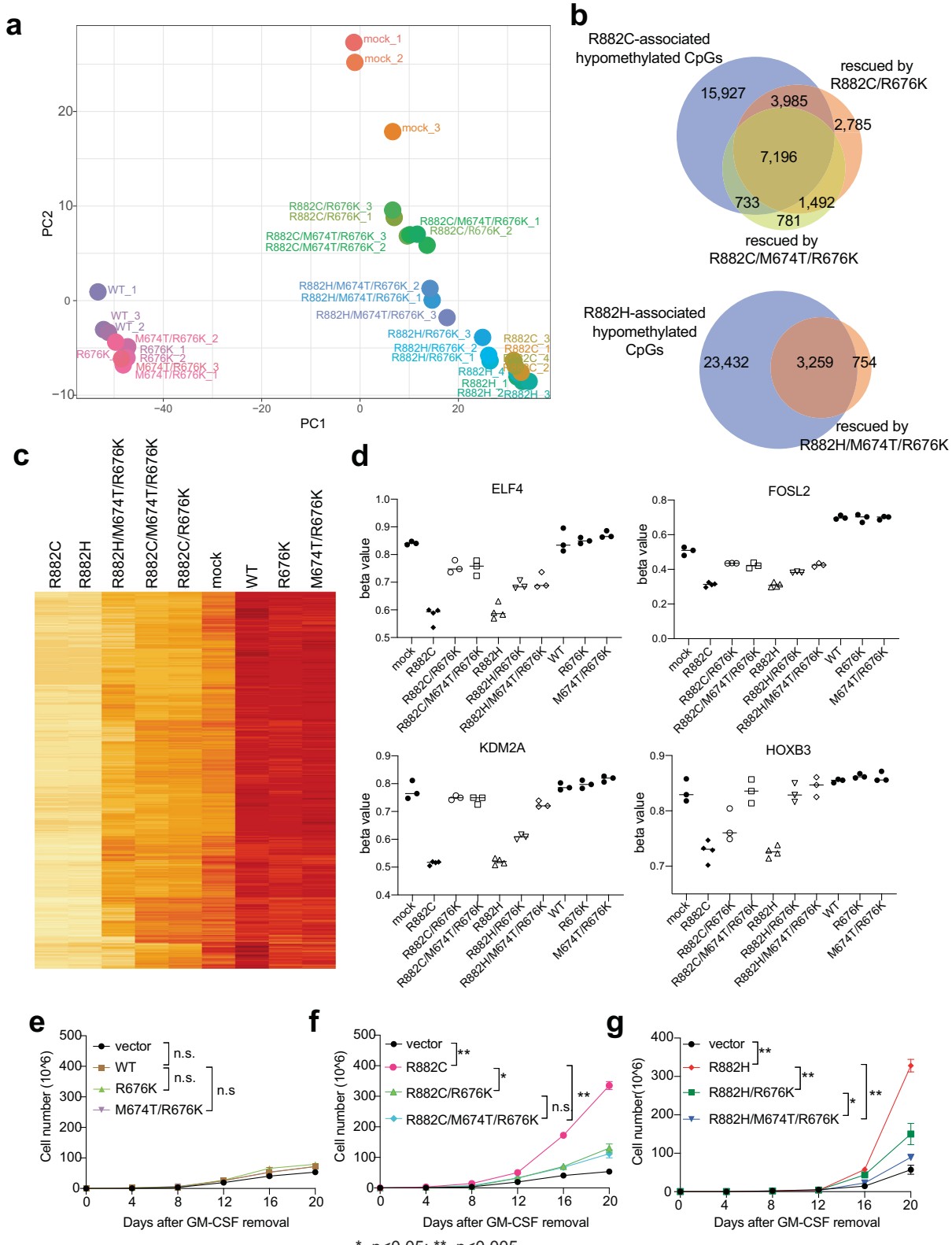

*, p<0.05; **, p<0.005

R882C-mutated DNMT3A proteins, those DNMT3A proteins carrying both disease mutation (R882H or R882C) and DNMT3B-converting mutation show reduced oligomerization propensity in vitro, leading to increased concentrations of DNMT3A particles in solution and consequently, enhanced chromatin targeting in cells. As a result, the DNA hypomethylation phenotype and cell proliferation resulting from the disease mutations were greatly suppressed by the rescuing mutations.

It remains to be investigated how this distinct oligomerization behavior of DNMT3A and DNMT3B adds to other structural and biochemical traits to orchestrate their distinct cellular activities, as well as their differential stability regulation by DNMT3L[57].

In summary, this study establishes a molecular basis for intervention of diseases involving DNMT3A hotspot mutations. The observation that a subtle change of the chemical environment at

**Fig. 6 | Rescue the DNMT3A hotspot mutation-induced CpG hypo-methylation and cytokine-independent growth in TF-1 cells. a** Principal component analysis (PCA) using the Infinium methylationEPIC array profiles of TF-1 cells with ectopic expression of the indicated DNMT3A, in comparison to vector control (mock). **b** Venn diagram using the differentially methylated CpGs (DMCs) exhibiting hypomethylation in TF-1 cells with a DNMT3A hotspot mutant (top: R882C; bottom: R882H) and those re-methylated by an additionally introduced mutation (R676K or M674T/R676K; △beta value greater than 10% and $q$ value less than 0.01 using paired $t$ test). **c** Heatmap showing the methylation level (beta values) at 20,888 DMCs that were hypomethylated upon transduction of a DNMT3A hotspot mutant when compared to mock. **d** DNA methylation levels (beta value in the $y$-axis) of

DMCs at EFL4, FOSL2, KDM2A, and HOXB3 in TF-1 cells with ectopic expression of the indicated DNMT3A. **e–g** Effect of the polymerization-attenuating mutation, either R676K or M674T/R676K, on the GM-CSF (cytokine)-independent proliferation of TF-1 cells, which were transduced with WT DNMT3A (**e**) or a hotspot mutant (R882C (**f**) or R882H (**g**)). The total cell numbers at day 20 post-removal of GM-CSF were used for paired $t$ test ($n = 3$ biological replicates). Data are mean ± SD. Statistical analysis used a two-tailed paired $t$ test. n.s, not significant ($p > 0.05$). **$p = 0.0013$ (R882C vs vector), **$p = 0.0056$ (R882C vs R882C/R676K), **$p = 0.0021$ (R882C vs R882C/M674T/R676K), **$p = 0.0045$ (R882H vs vector), **$p = 0.0066$ (R882H vs R882H/R676K), **$p = 0.0077$ (R882H vs R882C/M674T/R676K), and *$p = 0.0276$ (R882H/R676K vs R882H/M674T/R676K).

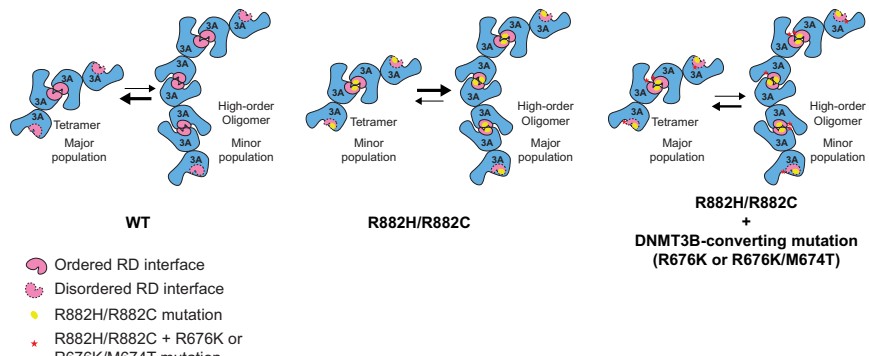

**Fig. 7 | A model for the dynamic assembly of DNMT3A, WT or mutant.** WT DNMT3A is dominated by the tetrameric assembly, whereas the R882H or R882C mutation shifts the equilibrium toward high-order oligomerization or even aggregation. Introducing the DNMT3B-converting mutation at the RD interface in part shifts the population of the assemblies back toward the tetramers.

the RD interface cascades strong protein polymerization raises the possibility of the development of a polymerization-attenuating agent through fine-tuning the conformations at the RD interface. These observations highlight the delicate balance of DNMT3A oligomerization in its functionality and more importantly, provide a conceptual framework for the development of small molecule inhibitors targeting the aberrant polymerization behavior of DNMT3A R882 mutations.

## Methods

### Plasmid
For cellular analysis, Myc-tagged human DNMT3A isoform 1 (DNMT3A1, referred to as DNMT3A herein) was cloned into MSCV-puro retroviral vector (Clontech) as previously described[39]. DNMT3A mutations were generated by QuikChange II XL site-directed mutagenesis kit (Agilent). For structural and biochemical characterizations, a DNA fragment encoding the human DNMT3A MTase domain (residues 628–912) or PWWP-ADD-MTase fragment (residues 281–912) was inserted into an in-house MBP vector, in which the inserted gene is preceded by an N-terminal hexahistidine (His$_6$)-MBP tag and a TEV cleavage site. All plasmids were verified by Sanger sequencing.

### Protein expression and purification
BL21(DE3) RIL cells harboring the expression plasmids were grown at 37 °C and induced by the addition of 0.13 mM (For MTase) and 0.067 mM (for PWWP-ADD-MTase) isopropyl β-D-1-thiogalactopyranoside when the cell density reached A$_{600}$ of 1.0. The cells continued to grow at 16 °C overnight, before being harvested and lysed in buffer containing 50 mM Tris-HCl (pH 8.0), 1 M NaCl, 25 mM imidazole, 10% glycerol, 10 µg/mL DNase I, and 1 mM PMSF. Subsequently, the fusion proteins were purified through a nickel column, followed by removal of the MBP tag by TEV cleavage, HiTrap Heparin HP chromatography and size-exclusion chromatography on a HiLoad 16/600 Superdex 200 pg column (GE Healthcare) in 20 mM Hepes-NaOH (pH 7.2), 250 mM NaCl, 5% glycerol and 5 mM DTT. The purified protein samples

were concentrated and stored at −80 °C for further use. The mutants were constructed by site-directed mutagenesis and purified in the same manner as described above.

### Crystallization and data collection
For crystallization, 10–13 mg/mL WT or ~17 mg/mL mutant (R882H/ R676K or R882C/R676K) DNMT3A MTase domain was each incubated with 0.2–0.4 M potassium sodium tartrate using the sitting drop vapor-diffusion method at 4 °C. After reaching sufficient size, the crystals were soaked in the well solution supplemented with 25% ($v/v$) glycerol before flash frozen in liquid nitrogen. X-ray diffraction data were collected on beamlines 5.0.2 and 5.0.3 at the Advanced Light Source, Lawrence Berkeley National Laboratory. The datasets were processed with the HKL3000 program[58]. The structure of the DNMT3A complex was solved by molecular replacement with PHASER[59] using DNMT3A/3L-DNA structure 5YX2 as a search model. Iterative cycles of model rebuilding and refinement were carried out using COOT[60] and PHENIX[61], respectively. The statistics for data processing and structure refinements are summarized in Supplementary Table 1.

### In vitro DNA methylation assay
Synthesized 36-bp (GAC)$_{12}$/(GTC)$_{12}$ DNA duplex (Integrated DNA Technologies) was annealed as a reaction substrate. For concentration-dependent DNA methylation assay, a 20-µL reaction mixture contained 0.75 µM DNA substrate, DNMT3A at various concentrations (0.375 µM, 0.75 µM, and 1.5 µM), 0.55 µM S-adenosyl-L-[methyl-$^3$H] methionine (specific activity 87.9 Ci/mmol, PerkinElmer), 0.65 µM AdoMet in 50 mM Tris-HCl (pH 7.5), 26 mM NaCl, 0.05% β-mercaptoethanol, 5% glycerol and 200 µg/mL BSA. The methylation assays were carried out in triplicate at 37 °C for 30 min and quenched by the addition of 5 µL of 10 mM AdoMet. For the steady-state kinetic assay, a 20-µL reaction mixture contained the DNA substrate at various concentrations (0, 0.02, 0.04, 0.1, 0.13, 0.16, and 0.24 µM), 0.2 µM DNMT3A WT or 0.4 µM DNMT3A mutant, 0.55 µM S-adenosyl-L-[methyl-$^3$H] methionine

(specific activity 87.9 Ci/mmol, PerkinElmer), 0.65 μM AdoMet in 50 mM Tris-HCl (pH 7.5), 26 mM NaCl, 0.05% β-mercaptoethanol, 5% glycerol, and 200 μg/mL BSA. For measurement of methylate rate, the DNA methylation assays were carried out in triplicate at 37 °C for 0 min, 10 min, and 20 min, respectively, before being quenched by the addition of 5 μL of 10 mM AdoMet. For detection, 8 μL of the reaction mixture was spotted on Amersham Hybond-XL paper (GE Healthcare) and dried out. The paper was then washed with 0.2 M cold ammonium bicarbonate (pH 8.2) (twice), Milli Q water, and Ethanol. Subsequently, the paper was air-dried and transferred to scintillation vials filled with 3 mL ScintiVerse cocktail (Thermo Fisher). The radioactivity of tritium was measured with a Beckman LS6500 counter. $K_m$ and $k_{cat}$ values were determined using the Michalis–Menten equation.

### Size-exclusion chromatography analysis of DNMT3A proteins

Purified DNMT3A or mutant proteins (100 μL of 3 mg/mL protein) were loaded onto a Superose 6 Increase 10/300 GL column (GE Healthcare) at 4 °C. The flow rate was kept constant at 0.5 mL/min using a running buffer of 20 mM Hepes (pH 7.2), 250 mM NaCl, 5% glycerol, and 5 mM DTT.

For comparison of the homo-oligomeric DNMT3A MTase domain with the hetero-tetrameric complex between DNMT3A MTase and DNMT3L C-terminal domain (residues 178–386), purified proteins (100 μL of 0.5 mg/mL protein) were loaded onto a Superdex™ 200 Increase 10/300 GL column (GE Healthcare), pre-equilibrated with buffer containing 20 mM Hepes (pH 7.2), 250 mM NaCl, 5% Glycerol and 5 mM DTT. The flow rate was kept constant at 0.3 mL/min.

### DLS

MBP-tagged DNMT3A MTase domain, WT or mutants, and DNMT3B MTase domain were diluted to 0.7 mg/mL in buffer containing 20 mM Hepes (pH 7.2), 250 mM NaCl, and 5 mM DTT. DLS of the samples was measured in a 384-well microwell plate (3820, Corning) by DynaPro Plate Reader II (Wyatt Technology Corporation) at 25 °C. The hydrodynamic radii were calculated using the DYNAMICS software (Wyatt Technology Corporation) and averaged with triplicate measurements.

### Thermal shift assay

To prepare the sample mixture, 1 μM WT or mutant DNMT3A PWWP–ADD-MTase protein was dissolved in a 20-μL buffer containing 20 mM HEPES (pH 7.2), 5% glycerol, 250 mM NaCl, 5 mM DTT, and 1× GloMelt Dye. The experiment was conducted using a BioRad CFX96 connect real-time PCR detection system, as previously described[36], with the sample plate subjected to stepwise heating from 4 to 95 °C at 0.5 °C per increment. Fluorescence intensity was recorded with the excitation and emission wavelength set to 470 nm and 510 nm, respectively. The experiments were performed in triplicate.

### Electrophoretic mobility shift assay

Each sample contained 0.1 μM (GAC)$_8$/(GTC)$_8$ DNA duplex mixed with 0, 0.1, 0.2, 0.4, 0.6, 0.8, or 1 μM MBP-tagged DNMT3A MTase domain, WT or mutant, dissolved in 10-μL binding buffer containing 20 mM Tris-HCl (pH 7.2), 100 mM NaCl, 5% glycerol and 5 mM DTT. Samples were incubated on ice for 30 min before being resolved on a 5% *w/v* polyacrylamide gel (59:1 for acrylamide:bis-acrylamide), which was run at 100 V using 0.2× TBE (pH 8.3) running buffer for 40 min at 4 °C. The gel was stained with SYBR™ Gold (Thermo Fisher Scientific) and visualized by ChemiDoc Imaging System (Bio-Rad).

### Cell culture and stable cell line generation

The human TF-1 erythroleukemic cell line (ATCC #CRL-2003) was cultured and maintained in the RPMI 1640 base medium (Invitrogen) supplemented with 10% of fatal bovine serum (VWR) and 2 ng/mL of recombinant human GM-CSF (Peprotech). MSCV-based retrovirus encoding WT or mutant DNMT3A was packaged in HEK293T cells (ATCC #CRL-3216) and used for infecting the parental TF-1 cells. 48 h post-infection, TF-1 cells were treated with 2 μg/mL of puromycin for seven days to generate stable expression cell lines, which were subsequently maintained in the culture medium added with 0.5 μg/mL of puromycin. GM-CSF removal and cytokine-independent growth were performed as previously described[39]. Briefly, 100,000 cells were seeded in triplicate per group to each well in the 24-well plates. The cytokine-independent growth of TF-1 cells was examined by counting the total number of viable cells by trypan blue exclusion assay at day 0, 4, 8, 12, 16, and 20 after removal of GM-CSF. Cells within 10 passages were used in the study.

### Quantifications of 5-mdC in genomic DNA by using LC-MS/MS/MS

The measurements of 5-mdC and 2′-deoxyguanosine (dG) in genomic DNA were conducted by following a previously published protocol[15,27]. Briefly, 1 μg of genomic DNA was digested with 0.1 unit nuclease P1 in a buffer of 30 mM sodium acetate (pH 5.6) and 1 mM ZnCl$_2$. After a 24-h incubation at 37 °C, to the mixture were added 0.5 units of alkaline phosphatase and 0.001 units of phosphodiesterase I in 50 mM Tris-HCl (pH 8.6). The mixture was incubated at 37 °C for another 2 h, and [$^{13}C_5$]-5-mdC and [$^{15}N_5$]-dG were added to the enzymatic digestion mixture of 50 ng of DNA. The enzymes were removed from the digestion mixture by chloroform extraction. The ensuing aqueous layer was dried using a Speed-vac and the dried residues were subsequently reconstituted in doubly distilled water. The digestion mixture of ~5 ng DNA was subjected to LC-MS/MS/MS analyses on an LTQ XL linear ion trap mass spectrometer (Thermo Fisher Scientific) equipped with a nanoelectrospray ionization source, where an EASY-nLC II system (Thermo Fisher Scientific) was employed for the separation. The amounts of 5-mdC and dG (in moles) in the nucleoside mixtures were calculated from area ratios of peaks found in selected-ion chromatograms for the analytes over their corresponding isotope-labeled standards, the amounts of the labeled standards added (in moles), and the calibration curves. The levels of 5-mdC, reported as the percentages of dG, were calculated by dividing the molar quantities of 5-mdC with those of dG.

### Infinium methylationEPIC array-based profiling of DNA methylation and data analysis

Total genomic DNA was extracted using the PureLink™ Genomic DNA Mini Kit (Invitrogen), followed by bisulfite conversion using the EpiTect Bisulfite Kit (Qiagen). DNA methylation profiling was performed by the UNC mammalian genotyping core using the Infinium methylationEPIC Kit v1.0 (Illumina) according to the manufacturer's instructions. Methylation data were then subject to background subtraction and control normalization minfi in R (version 1.40.0)[62]. PCA was performed using beta values of all samples calculated from the Infinium methylationEPIC array. Differentially methylated CpGs (DMCs) were identified using dmpFinder in a categorical mode. Methylation changes were considered significant using a cutoff of $q$ value of less than 0.01 and a difference of beta value of more than 10%. Hierarchical clustering analysis, scatter plots, and density plots were generated in R using "pheatmap," and "ggplot2" packages as described previously[40]. k-mer Probability logo analysis was performed using kpLogo (http://kplogo.wi.mit.edu/). The 20,888 overlapped R882-hypomethylated DMCs and 765 hypermethylated DMCs are used for this analysis. All 865,917 probes served as background.

### Cleavage under targets and tagmentation (CUT&Tag) and data analysis

CUT&Tag followed by deep sequencing was performed with a commercially available kit according to the manufacturer's instruction (EpiCypher CUTANA™ pAG-Tn5 for CUT&Tag, Cat# 15-1017). 1000 of NIH-3T3 cells (ATCC #CRL-1658) with stable expression of Myc-tagged WT DNMT3A were added to all TF-1 cell groups as spike-in

normalization control. Sequencing was conducted using an Illumina NextSeq 500 Sequencing platform (available from the core facility of UNC Pharmacology Department) as previously described[63]. Briefly, the .fastq files were mapped to the GRCh38 human genome using bowtie2.3.5[64]. Non-primary alignment, PCR duplicates, or blacklist regions were removed from aligned data by Samtools (v1.9)[65], Picard 'MarkDuplicates' function (v2.20.4) ("Picard Toolkit" 2019. Broad Institute, GitHub Repository. http://broadinstitute.github.io/picard/; Broad Institute), and bedtools (v2.28.0)[66], respectively. To calibrate CUT&Tag signals in TF-1 cells (human sample), the spike-in control genome (GRCm38) was used to quantitatively compare the genomic profiles, and the scale factor is calculated as below.

$$\text{Scale factor} = R_{\text{sample}} \times R_{\text{ctrl-spikein}} / R_{\text{sample-spikein}} \qquad (1)$$

$$R_{\text{sample}}: \text{reads of each sample mapped to GRCh38} \qquad (2)$$

$$R_{\text{ctrl-spikein}}: \text{reads of control (WT) sample mapped to GRCm38} \qquad (3)$$

$$R_{\text{sample-spikein}}: \text{reads of each sample mapped to GRCm38} \qquad (4)$$

Peak calling was performed using MACS2[67]. The distribution of peaks was calculated by the 'annotatePeaks.pl' function of HOMER (Hypergeometric Optimization of Motif Enrichment)[68]. Deeptools (v3.3.0) was used to produce the bigwig files and averaged plotting of CUT&Tag signals[69]. GREAT analysis was conducted as before[70].

## Reporting summary
Further information on research design is available in the Nature Portfolio Reporting Summary linked to this article.

## Data availability
All data needed to evaluate the conclusions in the paper are present in the paper and/or the supplementary materials. Coordinates and structure factors for homotetrameric DNMT3A^WT, DNMT3A^R882H/R676K, DNMT3A^R882C/R676K, and DNMT3A^R882H/N879A have been deposited in the Protein Data Bank under accession codes 8TDR, 8TE1, 8TE3, and 8TE4, respectively. The Cut&Tag and DNA methylation profiling data have been deposited in NCBI's Gene Expression Omnibus under GEO series accession number GSE225828 and GSE226062, respectively. Source data are provided as a Source Data file with this paper. Source data are provided with this paper.

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

## Acknowledgements

We thank staff members at the Advanced Light Source, Lawrence Berkeley National Laboratory for access to the X-ray beamline. The Berkeley Center for Structural Biology is supported in part by the National Institutes of Health, National Institute of General Medical Sciences, and the Howard Hughes Medical Institute. The Advanced Light Source is supported by the Director, Office of Science, Office of Basic Energy

Sciences, of the U.S. Department of Energy under Contract No. DE-AC02-05CH11231. The Pilatus detector was funded under NIH grant S10OD021832. The ALS-ENABLE beamlines are supported in part by the National Institutes of Health, National Institute of General Medical Sciences, grant P30 GM124169. This work was supported by NIH grants (R35GM119721 to J.S., R01 CA215284 and R01 CA211336 to G.G.W., and R35 ES031707 to Y.W.) and University of California Cancer Research Coordinating Committee (UC CRCC) grant (CRR-20-634140) to J.S. and G.G.W., is American Cancer Society (ACS) Research Scholar and a Leukemia & Lymphoma Society (LLS) Scholar.

## Author contributions

J.L., Y.G., J.Y. and J.C. performed experiments. Y.W., G.G.W. and J.S. organized the study. G.G.W. and J.S. wrote the manuscript with input from all the authors.

## Competing interests

The authors declare no competing interest.
