## [Peer Review File · Nature Communications]

Structure-guided functional suppression of AML-associated DNMT3A hotspot mutationsREVIEWER COMMENTS

Reviewer #1 (Remarks to the Author):

The manuscript by Lu J et al reported the structural and functional studies on the AML-associated DNMT3A hotspot mutation R882H and R882C. The combined structural, biochemical, and cell biology studies highlighted the important role of R882H and R882C in DNMT3A oligomerization and in reshaping the DNA methylome in cell. The story provides new insights into potential therapeutic strategies. However, some concerns are also raised which need more discussion or analysis.

Major:

Arg882 is located at the RD interface of the two central subunits but not involved in packing of external ones. So, the current structures only prove that the R882H or R882C could make the RD interface more stable but not create new interface for oligomerization. So, why the R882H or R882C can promote oligomerization? Where is the additional interface to mediate oligomerization? The author should explain the logic to link the RD interface stability to protein oligomerization.

Minor:

The canonical hydrogen bond only refers to hydroxyl group. The sulfhydryl group of C882 is too weak to form a hydrogen bond.

Fig 4b looks identical as 2d. Maybe remove it to avoid redundancy.

Reviewer #2 (Remarks to the Author):

Major Points

The data presented in this manuscript support the conclusion that mutations in RD interface of R882H DNMT3A mutant protein that increase the distance between the interface stabilize the tetrameric structure and increase the soluble active fraction of the enzyme. Although some aspects of this study are novel, several gaps and missing experiments are needed to support the arguments. All the experiments done in R882H or R882C mutants would need a comparable WT control to draw conclusions based on rescue effects and distinguish them from gain of function effect. Conclusions regarding change in DNA binding properties need either cocrystal structure or minimum DNA binding assays. Activity assays to determine catalytic constants must be done as time course reactions and rates fit MM kinetics to determine the constants.

Please make a clear distinction between oligomerization and aggregation.

Please see the details below.

Data Figure 1, although it provides the first report of the Dnmt3A homotetrameric structure, there is very little new information since it is similar to the DNMT3A 3L structure. The external subunits are disordered. The disordered external subunits could result from the absent RD interface, which stabilizes the structure of the central dimer. This observation also leads to a potential for Dnmt3A to oligomerize at the RD interface. This oligomerization in the presence of long DNA substrates was shown to increase the enzyme activity by cooperativity by several labs in multiple publications, which has not been discussed here. Visualizing external subunits in the presence of DNA would be more informative since many residues in the RD interface interact with DNA. It is possible that external subunits are more ordered in the presence of DNA. This suggestion is supported by the previous cocrystal structure of DNMT3A WT/3L and DNMT3AR882H/3L with DNA

Oligomerization is often referred to as the ordered structural property of an active enzyme, while aggregation results in precipitation and loss of activity. The authors mix up the two properties, and it is unclear what they refer to by oligomerization. DNMT3A forms large aggregates in solution, which elute almost in the void volume, and also forms soluble ordered homo or heterotetramers. AML mutant has lost the ability to make soluble tetramers and only forms large aggregates, referred to in the manuscript as oligomers, except in a few places, such as line 225, which correctly points it out as aggregation behavior. Since DNMT3A can cooperatively bind to DNA, as shown by EMSA and catalytic activity assays by Jeltsch, Gowher, and Reich Group, it would be essential to discuss the oligomer formation in the presence and absence of DNA.

Norvil et al. Biochemistry also showed that the R823H of Dnmt3b does not affect its activity. It showed slightly higher activity than WT and suggested a potentially different RD interface in DNMT3B, which fits the data in Fig 2e.

It is also essential to check the effect of DNMT3B converting mutations on the WT DNMT3A enzyme structure and activity. Given that AML Arg is the common residue between DNMT3A and 3B, it is unclear why the rescue by DNMT3B converting mutations partially rescues DNMT3A activity even though their structure based on chromatograms is almost completely rescued. A cocrystal structure with DNA or DNA binding assays showing the dominant defect of substrate binding in AML mutants should be provided to support the results.

The activity assay in Figure 4d is almost saturating at .8uM. It is unclear if the points represent the rates of reaction at different substrate concentrations or if this assay is a single time point with different substrate concentrations. There is only one point on the linear region of the graph and therefore K_m cannot be accurately calculated. Please repeat the assays with lower concentrations of DNA and the time courses for each substrate concentration should be shown in the supplementary data.

It would also be interesting to determine if the rescued hypomethylated CpGs in Figure 6b are also targets of DNMT3B. Given that DNMT3A and 3B have redundant targets and a recent publication (Norvil et al. NAR) showed that R882H mutant has a flanking sequence preference similar to DNMT3A and that R882H mutant rescued DNMT3B targets in DKO ESCs. How does the ectopic expression of DNMT3B compare to R882H and R882H plus DNMT3B converting mutations?

Of the 20,888 hypomethylated sites with A or T at the N+1 flank, how many were rescued by DNMT3B converting mutations? What fraction of these is represented in Figure 6b

The statement in the abstract, “The hotspot DNMT3A mutations at the site of Arg822 (R882) promote high-order oligomerization, leading to aberrant DNA methylation that contributes to the pathogenesis of acute myeloid leukemia (AML)” is potentially misleading. The AML mutant does form aggregates and has lost the ability to make soluble tetramers, which elute later. However, what is the evidence that these aggregates or oligomers are directly responsible or contribute to the pathogenesis? If there are, please cite. Otherwise, please modify the statement so it doesn’t indicate direct dependence.

Reviewer #3 (Remarks to the Author):

This manuscript reports the homo-tetrameric crystal structure of the MTase domain of DNMT3A as well as the mutants bearing the acute myeloid leukemia (AML)-associated R882 mutations. The crystal structure comparison shows that mutations of R882 to H or C increases the interactions between the two central DNMT3A molecules within the homo-tetramer. Combining with biochemical and genomic DNA methylation results, the authors conclude that a molecular basis is established for the high-order oligomer formation and reduced methylation activities by R882H and R882C mutations in DNMT3A. The crystal structures of the wild type and R882H mutant of DNMT3A-DNMT3L tetramer in complex with DNA have already been published (Nat. Comm. 2020), revealing a reduced DNA binding at the protein-DNA interface and dynamic TRD loop to explain the hypomethylation caused by the hotspot AML-associated R882H mutation in DNMT3A. This manuscript provides incremental information for the additional interactions within DNMT3A homodimer caused by the mutations. Some major concerns for the interpretation of the DNMT3A tetrameric structures are listed below.

1. DNMT3A forms a homo-tetramer and R882 is located in the RD interface between two central DNMT3A molecules. The crystal structures of the double mutants reveal that the mutated H882 and C882 engage in dimeric interactions in the RD interface but the wild-type R882 does not make any intermolecular contacts. The authors thus suggest that R882H and R882C mutations increase intermolecular interactions and therefore promote DNMT3A oligomerization. However, the R882H and R882C mutations enhance the intermolecular interactions between the two central DNMT3A molecules within the homo-tetramer, and they are not involved in making interactions outside of the homo-tetramer. So based on the crystal structures, these mutations likely stabilize the homo-tetramer assembly, but not promote high-order oligomer formation
2. It is not clear if the oligomerization of DNMT3A is linked to the decreased methylation activity and onset of disease. What is the proposed mechanism of mutation-induced oligomerization affecting the activity of DNMT3A?
3. The stability or folding of full-length DNMT3A mutants should be tested because the decreased activity could be resulted from mutation-induced instability of protein.

4. In this manuscript, in vitro activity assays show that wild-type and mutated DNMT3A exhibit similar K_m values (Figure 4d), suggesting similar DNA-binding affinities. But the cell-based studies reveal that DNMT3A R882 mutations reduce chromatin binding and result in hypo-methylation (Figure 5 and 6). These are contradictory results.

5. The definition of “gain of function” by mutation for DNMT3A mutants is mis-interpreted in this manuscript. The authors state in the Abstract and Introduction that R882 mutations led to attenuated DNA methylation by WT DNMT3A in mouse ES or TF1 leukemia cell line, suggesting a gain-of-function effect of these mutations. Since the mutations of R882 produce DNMT3A mutants with reduced enzymatic activities, these mutations cause a “loss-of-function” effect but not a gain-of-function effect

General Response

We thank all three reviewers for their collective efforts in reviewing our manuscript, their positive view of our work, and their constructive comments to improve our manuscript. As outlined below in the point-by-point response (marked in blue), we have now systematically addressed all the raised critiques and have incorporated them in the revised manuscript (marked in red).

Point-by-point Response

Response to Reviewer 1

Reviewer #1 (Remarks to the Author):

The manuscript by Lu J et al reported the structural and functional studies on the AML-associated DNMT3A hotspot mutation R882H and R882C. The combined structural, biochemical, and cell biology studies highlighted the important role of R882H and R882C in DNMT3A oligomerization and in reshaping the DNA methylome in cell. The story provides new insights into potential therapeutic strategies. However, some concerns are also raised which need more discussion or analysis.

We greatly appreciate the reviewer for pointing out that “The combined structural, biochemical, and cell biology studies highlighted the important role of R882H and R882C in DNMT3A oligomerization and in reshaping the DNA methylome in cell” and “The story provides new insights into potential therapeutic strategies”. In the revised manuscript, we have addressed the major and minor concerns raised by the reviewer, as summarized below.

Major:

Arg882 is located at the RD interface of the two central subunits but not involved in packing of external ones. So, the current structures only prove that the R882H or R882C could make the RD interface more stable but not create new interface for oligomerization. So, why the R882H or R882C can promote oligomerization? Where is the additional interface to mediate oligomerization? The author should explain the logic to link the RD interface stability to protein oligomerization.

We thank the reviewer for raising this point and apologize for the lack of clarity in our original manuscript. We wish to clarify that, while our crystallographic study captured a homotetrameric state of DNMT3A, it is most likely that DNMT3A in solution undergoes a dynamic transition between various assembly states. In support of this notion, our recent cryo-EM study reveals that whereas the homotetrameric assembly represents the major form of DNMT3B, it can further associate into a hexameric form or dissociate into a trimeric form (Ref. 36). To further strengthen this point, we compared the size-exclusion chromatography result of homo-oligomeric DNMT3A with that of the stable, heterotetrameric DNMT3A-DNMT3L complex in the revised manuscript (Fig. R1a,b). The homo-oligomeric DNMT3A eluted in a much broader peak than DNMT3A-DNMT3L, supporting that DNMT3A, like DNMT3B, is under a dynamic equilibrium between homotetramer and other oligomeric states in solution. In this context, stabilization of the RD

interface by the R882H or R882C mutation promotes the oligomerization process, thereby facilitating the transition of DNMT3A or DNMT3B into polymeric forms (Fig. R1c). On the other hand, introducing additional R676K/M674T mutations helps reverse the oligomerization-promoting effect of the R882H or R882C mutation through fine-tuning the RD interface (Fig. R1c). We have included the new data and discussion in the revised manuscript.

Figure R1. Dynamic assembly of DNMT3A in solution. (a) Size-exclusion chromatography analysis of the C-terminal MTase domain of DNMT3A (3Ac) in homo-oligomeric form (green) or in complex with the C-terminal domain of DNMT3L (3Ac-3Lc) (blue). (b) SDS-PAGE images of corresponding fractions in (a), with elution volume marked. (c) A model for the dynamic assembly between homotetramer and higher-order oligomer of DNMT3A. Wild type (WT) DNMT3A is dominated by homotetramer, whereas the R882H or R882C mutation shifts the equilibrium toward polymer. Introducing the polymerization-attenuating mutation R676K or R676K/M674T partially reverses the equilibrium toward tetramer. These data have been included as Supplementary Fig. 5a,b and 9 in the revised manuscript.

Minor:

The canonical hydrogen bond only refers to hydroxyl group. The sulfhydryl group of C882 is too weak to form a hydrogen bond.

We thank the reviewer for the comment. Our consideration of assigning a hydrogen bond to the side chain of C882-mediated interaction was based on several theoretical investigations, including a previous report by Mazmanian et al (PMID: 27635780). To alleviate the reviewer's concern, we clarified that the sulfhydryl group of C882-mediated interaction is weak in the revised manuscript. (Line 249)

Fig 4b looks identical as 2d. Maybe remove it to avoid redundancy.

We thank the reviewer for the comment. We wish to clarify that Fig. 2d involves a long C-terminal fragment of DNMT3A, comprised of the PWWP, ADD and MTase domains, whereas Fig. 4b involves only the MTase domain of DNMT3A. Considering that the N-terminal domains of DNMT3A or DNMT3B may also modulate its oligomeric state, as illustrated for DNMT3B in our recent work (Ref 36), we keep both figures for proper structure-function correlation.

Response to Reviewer 2

Reviewer #2 (Remarks to the Author):

Major Points

The data presented in this manuscript support the conclusion that mutations in RD interface of R882H DNMT3A mutant protein that increase the distance between the interface stabilize the tetrameric structure and increase the soluble active fraction of the enzyme. Although some aspects of this study are novel, several gaps and missing experiments are needed to support the arguments. All the experiments done in R882H or R882C mutants would need a comparable WT control to draw conclusions based on rescue effects and distinguish them from gain of function effect. Conclusions regarding change in DNA binding properties need either cocrystal structure or minimum DNA binding assays. Activity assays to determine catalytic constants must be done as time course reactions and rates fit MM kinetics to determine the constants. Please make a clear distinction between oligomerization and aggregation.

We thank the reviewer for point out that our data support “the conclusion that mutations in RD interface of R882H DNMT3A mutant protein that increase the distance between the interface stabilize the tetrameric structure and increase the soluble active fraction of the enzyme”. We also apologize for the lack of clarity regarding the relationship between DNMT3A oligomerization and its DNA methylation activity in our original manuscript. In the revised manuscript, we have included additional data and discussion to address the reviewer's concerns.

Please see the details below.

Data Figure 1, although it provides the first report of the Dnmt3A homotetrameric structure, there is very little new information since it is similar to the DNMT3A 3L structure. The external subunits are disordered. The disordered external subunits could result from the absent RD interface, which stabilizes the structure of the central dimer. This observation also leads to a potential for Dnmt3A to oligomerize at the RD interface. This oligomerization in the presence of long DNA substrates was shown to increase the enzyme activity by cooperativity by several labs in multiple publications, which has not been discussed here. Visualizing external subunits in the presence of DNA would be more informative since many residues in the RD interface interact with DNA. It is possible that external subunits are more ordered in the presence of DNA. This suggestion is supported by the previous cocrystal structure of DNMT3A WT/3L and DNMT3A R882H/3L with DNA.

We thank the reviewer for the comment on Figure 1. We wish to emphasize that the novelty of this study lies not only in the fact that it provides the first report of the DNMT3A homotetrameric structure, but also in that it provides a framework for understanding the impact of the AML hotspot mutation on DNMT3A oligomerization, therefore providing mechanistic insights into therapeutic intervention against DNMT3A R882H/R882C-related diseases.

We especially thank the reviewer for raising the point that the cooperative binding of DNMT3A oligomer to DNA increases its DNA methylation efficiency, as demonstrated by Jeltch, Gowher and Reich groups, which highlights the importance of maintaining a balanced assembly equilibrium in DNMT3A functionalities. We also agree that DNA binding may lead to structural ordering of the RD interface, and consequently, increased oligomerization of DNMT3A. We have included these points and have cited the related works in the discussion of the revised manuscript (line 476-484).

Oligomerization is often referred to as the ordered structural property of an active enzyme, while aggregation results in precipitation and loss of activity. The authors mix up the two properties, and it is unclear what they refer to by oligomerization. DNMT3A forms large aggregates in solution, which elute almost in the void volume, and also forms soluble ordered homo or heterotetramers. AML mutant has lost the ability to make soluble tetramers and only forms large aggregates, referred to in the manuscript as oligomers, except in a few places, such as line 225, which correctly points it out as aggregation behavior. Since DNMT3A can cooperatively bind to DNA, as shown by EMSA and catalytic activity assays by Jeltsch, Gowher, and Reich Group, it would be essential to discuss the oligomer formation in the presence and absence of DNA.

The reviewer's point is well taken. We apologize for the mixed use of terms in our original manuscript. In the revised manuscript, we have included the discussion of DNMT3A oligomer formation in the presence and absence of DNA (line 476-484). We also wish to clarify that our structural study indicates that formation of the large DNMT3A R882H/ R882C particles, which presumably remain structurally ordered, follows a similar molecular mechanism as the formation of homotetramer. As supported by our DNA methylation analysis, the activity loss of DNMT3A R882H might in part be attributed to the polymerization of DNMT3A, which led to a reduction of the effective concentration of DNMT3A particles for genomic targeting. As the reviewer and other

experts (such as Carl Frieden, PMID 17962399) correctly pointed out, aggregation implies a non-specific process whereas oligomerization/polymerization implies a defined process, we refer to the large-sized DNMT3A R882H/R882C particles as polymers and the process as polymerization wherever suitable in the revised manuscript. We are open to further modification of these terms as the reviewer deems appropriate.

Norvil et al. Biochemistry also showed that the R823H of Dnmt3b does not affect its activity. It showed slightly higher activity than WT and suggested a potentially different RD interface in DNMT3B, which fits the data in Fig 2e. It is also essential to check the effect of DNMT3B converting mutations on the WT DNMT3A enzyme structure and activity.

We thank the reviewer for raising this excellent point and have included a discussion on the previous work by Norvil in the revised manuscript. Following the reviewer's suggestion, we have examined the oligomerization state and DNA methylation activity of the DNMT3B-converting mutants. As shown in Fig. R2, The DNMT3B-converting mutations R676K and R676K/M674T modestly dampened the DNMT3A oligomerization (Fig. R2a), leading to a slight increase in their DNA methylation activity. We have included this new data in the revised manuscript.

Figure R2. Biochemical characterization of DNMT3B-converting mutations. (a) Size-exclusion chromatography analysis of the MBP-tagged MTase domain of DNMT3A, WT (black), R676K (green) and M674T/R676K (blue). Elution volumes for select molecular weight standards were marked. (b) *In vitro* DNA methylation assays for DNMT3A MTase domain, either WT, R676K or M674T/R676K mutant. Statistical analysis used two-tailed Student's t-test. ns, not significant; *, $p < 0.05$. These data have been included as Supplementary Fig. 3c,d in the revised manuscript.

Given that AML Arg is the common residue between DNMT3A and 3B, it is unclear why the rescue by DNMT3B converting mutations partially rescues DNMT3A activity even though their structure based on chromatograms is almost completely rescued. A cocrystal structure with DNA or DNA binding assays showing the dominant defect of substrate binding in AML mutants should be provided to support the results.

We thank the reviewer for the comment. We wish to clarify that the decrease of DNA methylation activity of DNMT3A R882H was also in part attributed to the loss of the positive charge at site 882, as demonstrated by our previous study on the DNMT3A R882H-DNMT3L tetramer (Ref. 32) and Fig. R3. We have clarified this point in the revised manuscript (line 315-319).

Following the reviewer's comment on the defect of substrate binding in AML mutant, we performed EMSA experiments for MBP-tagged DNMT3A MTase with increasing concentration of a 24-bp DNA. The result shows that R882H mutation has a much lower DNA binding affinity compared to WT protein (Figure R4a, 4b), whereas the DNMT3B-converting mutations increase the DNA binding affinity to a modest extent (Figure R4c, 4d). These data have been included in the revised manuscript.

Figure R3. The EMSA analysis of the DNMT3A-DNA binding, WT or mutant. (a-d) a 24-bp (GAC)₈ DNA duplex was incubated with increased concentration of MBP-tagged MTase domain of WT (a), DNMT3A^{R882H} (b), DNMT3A^{R882H/R676K} (c), and DNMT3A^{R882H/M674T/R676K} (d). The experiment was performed twice with consistent results. These data have been included as Supplementary Fig. 6e-h in the revised manuscript.

The activity assay in Figure 4d is almost saturating at .8uM. It is unclear if the points represent the rates of reaction at different substrate concentrations or if this assay is a single time point with different substrate concentrations. There is only one point on the linear region of the graph and therefore Km cannot be accurately calculated. Please repeat the assays with lower

concentrations of DNA and the time courses for each substrate concentration should be shown in the supplementary data.

Following the reviewer's suggestions, we measured DNA methylation kinetics for DNMT3A WT and mutants, with multiple data points at lower DNA concentrations (Fig. R4). The result, consistent with our previous observation, support that the DNMT3B-converting mutation partially restored the DNA methylation activity of DNMT3A through increasing the effective concentration of DNMT3A particles.

Figure R4. *In vitro* DNA methylation analysis of WT and mutant DNMT3A. (a-d) DNA methylation kinetics for MBP-tagged DNMT3A MTase domain, WT (a), R882H/R676K (b), R882H/R676K/M674T (c), or R882H mutant (d). (e) Steady-state enzymatic kinetics of WT and mutant DNMT3A MTase domain, with the data derived from the 30-min measurement in (a-d). Data are mean \pm s.d. (n = 3 biological replicates). These data have been included in Fig. 4b and Supplementary Fig. 6a-d in the revised manuscript.

It would also be interesting to determine if the rescued hypomethylated CpGs in Figure 6b are also targets of DNMT3B. Given that DNMT3A and 3B have redundant targets and a recent publication (Norvil et al. NAR) showed that R882H mutant has a flanking sequence preference similar to DNMT3A and that R882H mutant rescued DNMT3B targets in DKO ESCs. How does the ectopic expression of DNMT3B compare to R882H and R882H plus DNMT3B converting mutations?

We appreciate the reviewer for this comment. As the reviewer may well know, the DNMT3B expression level in leukemia cells is generally low compared to that of DNMT3A. Also, unlike what was seen in ES cells, cancer cells express several enzymatically-inactive forms of DNMT3B that do not have enzymatic activity (ref 28-29). Therefore, the function of DNMT3B isoforms is rather complex in leukemia cells, an interesting point as reviewer 2 pointed out that warrants a future investigation.

As the reviewer 2 correctly pointed out, the previous study by Norvil et al has demonstrated that there exists similarity between DNMT3B and R882-mutated DNMT3A. In this study, our genomic DNA methylation analysis revealed that the R882C-associated hyper-methylated CpGs were enriched with purine at the N+1-flanking site, supporting the notion raised by Norvi et al on the enzymatic specificity of DNMT3A R882 mutant. On the other hand, In TF1 leukemia cells, ectopic expression of DNMT3B did not cause transformation (data not shown) as seen with R882H-mutated DNMT3A, suggesting a functional distinction between the two in cancer cells (although there is a degree of resemblance as the reviewer pointed out). Dissecting both similarity and distinctive features of the two (ie. DNMT3B and R882-mutated DNMT3A) merits a more detailed investigation in future. Following the reviewer's comment, we have included additional discussion on the findings by Norvil et al in the revised manuscript (Line 413-417).

Of the 20,888 hypomethylated sites with A or T at the N+1 flank, how many were rescued by DNMT3B converting mutations? What fraction of these is represented in Figure 6b

We have conducted the requested analysis. As shown in Fig. R5, out of the 7196 hypomethylated sites that were rescued by both of the DNMT3B-converting mutations R676K and M674T/R676K (Fig. 6b), 27.3% of the hypo-methylated sites with T at N+1 flank was rescued by DNMT3B-converting mutation, and 23.1% of the hypo-methylated sites with A at N+1 flank was rescued by DNMT3B-converting mutation. Therefore, we did not observe a dramatic difference at the two groups of sites regarding degree of rescue by DNMT3B converting mutations, consistent with the notion that the DNMT3B-converting mutations at the RD interface do not further impact the substrate specificity of the hotspot mutant of DNMT3A. The caveat of this analysis is the limited data size.

Figure R5. Flanking sequence analysis of the DNA methylation sites rescued by DNMT3B-converint mutation.

The statement in the abstract, “The hotspot DNMT3A mutations at the site of Arg822 (R882) promote high-order oligomerization, leading to aberrant DNA methylation that contributes to the pathogenesis of acute myeloid leukemia (AML)” is potentially misleading. The AML mutant does form aggregates and has lost the ability to make soluble tetramers, which elute later. However, what is the evidence that these aggregates or oligomers are directly responsible or contribute to the pathogenesis? If there are, please cite. Otherwise, please modify the statement so it doesn’t indicate direct dependence.

We thank the reviewer for the comment. While a couple of previous studies have linked the polymerization of DNMT3A R882H to its reduced DNA methylation activity and/or disease progression (Refs. 40 & 42), no direct evidence that DNMT3A polymerization is directly responsible for the pathogenesis. Following the reviewer’s comment, we have modified the statement to “The hotspot DNMT3A mutations at the site of Arg822 (R882) promote polymerization, leading to aberrant DNA methylation that may contribute to the pathogenesis of acute myeloid leukemia (AML)”.

Response to Reviewer 3

Reviewer #3 (Remarks to the Author):

This manuscript reports the homo-tetrameric crystal structure of the MTase domain of DNMT3A as well as the mutants bearing the acute myeloid leukemia (AML)-associated R882 mutations. The crystal structure comparison shows that mutations of R882 to H or C increases the interactions between the two central DNMT3A molecules within the homo-tetramer. Combining with biochemical and genomic DNA methylation results, the authors conclude that a molecular basis is established for the high-order oligomer formation and reduced methylation activities by R882H and R882C mutations in DNMT3A. The crystal structures of the wild type and R882H mutant of DNMT3A-DNMT3L tetramer in complex with DNA have already been published (Nat. Comm. 2020), revealing a reduced DNA binding at the protein-DNA interface and dynamic TRD loop to explain the hypomethylation caused by the hotspot AML-associated R882H mutation

in DNMT3A. This manuscript provides incremental information for the additional interactions within DNMT3A homodimer caused by the mutations. Some major concerns for the interpretation of the DNMT3A tetrameric structures are listed below.

We thank the reviewer for the summary of our work. We wish to emphasize that the novelty of this study lies not only in the fact that it provides the first report of the DNMT3A homotetrameric structure, but also in that it provides a framework for understanding the impact of the AML hotspot mutation on DNMT3A oligomerization, therefore providing mechanistic insights into therapeutic intervention against DNMT3A R882H/R882C-related diseases. In the revised manuscript, we have addressed the reviewer's concerns as follow.

1. DNMT3A forms a homo-tetramer and R882 is located in the RD interface between two central DNMT3A molecules. The crystal structures of the double mutants reveal that the mutated H882 and C882 engage in dimeric interactions in the RD interface but the wild-type R882 does not make any intermolecular contacts. The authors thus suggest that R882H and R882C mutations increase intermolecular interactions and therefore promote DNMT3A oligomerization. However, the R882H and R882C mutations enhance the intermolecular interactions between the two central DNMT3A molecules within the homo-tetramer, and they are not involved in making interactions outside of the homo-tetramer. So based on the crystal structures, these mutations likely stabilize the homo-tetramer assembly, but not promote high-order oligomer formation.

The reviewer's point is well taken. Reviewer 1 raises a similar point. We apologize for the lack of clarity in our original manuscript. As in our response to Reviewer 1, we have included the following data and discussion to clarify the effect DNMT3A R882H and R882C mutations on DNMT3A oligomerization.

First of all, our structural and biochemical characterizations of DNMT3A and DNMT3B have indicated that these two closely-related homologues undergo a dynamic transition between various assembly states in solution. As shown in our recent cryo-EM study of DNMT3B (Ref. 36), whereas DNMT3B mainly exists in the homotetrameric form, it can further associate into a hexameric form or dissociate into a trimeric form. Furthermore, we included new size-exclusion chromatography comparison of homo-oligomeric DNMT3A and the heterotetrameric DNMT3A-DNMT3L complex, which reveals a much broader elution peak for DNMT3A than that for DNMT3A-DNMT3L, therefore supporting that notion that DNMT3A, like DNMT3B, undergoes a dynamic transition between homotetramer and other alternative oligomeric states in solution. In this context, stabilization of the RD interface by the R882H or R882C mutation promotes the oligomerization process, thereby facilitating the transition of DNMT3A or DNMT3B into higher-order oligomeric forms (from tetramer to hexamer and beyond) (Fig. R1c). On the other hand, introducing additional R676K/M674T mutations helps reverse the polymerization-promoting effect of the R882H or R882C mutation through fine-tuning the RD interface (Fig. R1c). We have included the new data and discussion in the revised manuscript.

2. It is not clear if the oligomerization of DNMT3A is linked to the decreased methylation activity and onset of disease. What is the proposed mechanism of mutation-induced oligomerization affecting the activity of DNMT3A?

We apologize for the confusion. Previous studies by Russler-Germain et al (Ref. 40) and Nguyen et al (Ref. 42) have demonstrated that DNMT3A R882H enhanced DNMT3A oligomerization, resulting in reduced DNA methylation activity in vitro and in cells. Here, this study showed that compared with WT DNMT3A, the R882H and R882C mutation introduced new intermolecular interactions at the RD interface, providing a molecular explanation to the polymerization-promoting effect of these mutations. Based on these data, we propose that the enhanced oligomerization led to reduced effective concentrations of DNMT3A particles in cells, thereby causing disease-related DNA hypomethylation. We have clarified this point in our revised manuscript (Line 479-482).

3. The stability or folding of full-length DNMT3A mutants should be tested because the decreased activity could be resulted from mutation-induced instability of protein.

We thank the reviewer for the suggestions. As suggested, we focused on the DNMT3A2 fragment PWWP-ADD-MTase fragment, corresponding to the full-length DNMT3A2 isoform that is overexpressed in AML, and performed the thermal shift assays. Our result shows that WT and mutants of DNMT3A have nearly identical melting temperatures, thereby ruling out the possibility that decrease of enzymatic activity is due to mutation-induced instability of protein (Figure R6a-d).

Figure R6. Thermal shift assay for WT or mutant DNMT3A. (a) SDS-PAGE images of DNMT3A PWWP-ADD-MTase fragments, wild-type (WT) and mutants, used for the thermal shift assay. (b-d) Comparison of the first-order derivatives of the raw fluorescence data for WT DNMT3A with that of R882H or R882C mutant (b), R882H/R676K or R882H/R676K/M674T mutant (c), and R882C/R676K or R882C/R676K/M674T mutant (d). This data has been included as Supplementary Fig. 5d-g in the revised manuscript.

4. In this manuscript, in vitro activity assays show that wild-type and mutated DNMT3A exhibit similar K_m values (Figure 4d), suggesting similar DNA-binding affinities. But the cell-based

studies reveal that DNMT3A R882 mutations reduce chromatin binding and result in hypomethylation (Figure 5 and 6). These are contradictory results.

We thank the reviewer for the suggestions. As the reviewer may well know, the K_m value depends not only on the enzyme-substrate dissociation constant (K_d), but also on the turn-over rate K_{cat} , which makes it difficult to compare the DNA binding affinities directly. To clarify the mutational effect on DNA-binding affinity, we performed EMSA experiments by titrating a 24-mer DNA duplex with increasing concentration of DNMT3A proteins (Fig. R3). The result shows that the R882H mutation led to reduced DNA binding affinity (Figure R4a, b), consistent with the cellular assay. We also wish to clarify that the reduced chromatin binding by DNMT3A R882 mutations is conceivably caused by (i) impaired DNA interaction at the RD interface, as shown in our previous work (Ref. 32), and (ii) decreased concentrations of DNMT3A particles in solution. The DNMT3B-converting mutation increased the concentration of DNMT3A particles via attenuating the polymerization-promoting effect of DNMT3A R882H/R882C, thereby enhancing the chromatin binding of DNMT3A in cells. We have clarified this point in the revised manuscript (Line 318-322).

5. The definition of “gain of function” by mutation for DNMT3A mutants is mis-interpreted in this manuscript. The authors state in the Abstract and Introduction that R882 mutations led to attenuated DNA methylation by WT DNMT3A in mouse ES or TF1 leukemia cell line, suggesting a gain-of-function effect of these mutations. Since the mutations of R882 produce DNMT3A mutants with reduced enzymatic activities, these mutations cause a “loss-of-function” effect but not a gain-of-function effect.

We thank the reviewer for the comment. In the revised manuscript, we have replaced the term “gain of function” to “dominant-negative”.

REVIEWERS' COMMENTS

Reviewer #1 (Remarks to the Author):

My concerns have been addressed.

Reviewer #2 (Remarks to the Author):

All the major concerns have been satisfactorily addressed.

Reviewer #3 (Remarks to the Author):

The authors have addressed my questions and concerns.

The revised manuscript more clearly explains how the R882 mutations in the RD interface of DNMT3A tetramer may affect protein polymerization by the newly added Supplementary Figure 9. This figure is suggested to be moved to the main text of the manuscript.

Reviewer #1 (Remarks to the Author):

My concerns have been addressed.

Response: We thank the reviewer for his/her comment.

Reviewer #2 (Remarks to the Author):

All the major concerns have been satisfactorily addressed.

Response: We thank the reviewer for his/her comment.

Reviewer #3 (Remarks to the Author):

The authors have addressed my questions and concerns.

The revised manuscript more clearly explains how the R882 mutations in the RD interface of DNMT3A tetramer may affect protein polymerization by the newly added Supplementary Figure 9. This figure is suggested to be moved to the main text of the manuscript.

Response: We thank the reviewer's suggestion and have moved Supplementary Fig.9 to Figure 7 in the revised manuscript.